# Individual differences of limitation to extract beat from Kuramoto coupled oscillators: Transition from beat-based tapping to frequent tapping with weaker coupling

Nolan Lem[1]*, Takako Fujioka[1,2]

**1** Center for Computer Research in Music and Acoustics (CCRMA), Department of Music, Stanford University, Stanford, California, United States of America, **2** Wu Tsai Neurosciences Institute, Stanford University, Stanford, California, United States of America

* nlem@ccrma.stanford.edu

## Abstract

Musical performers synchronize to each other despite differences in sound-onset timings which reflect each musician's sense of the beat. A dynamical system of Kuramoto oscillators can simulate this spread of onsets at varying levels of temporal alignment with a variety of tempo and sound densities which also influence individual abilities for beat extraction. Here, we examined how people's sense of beat emerges when tapping with Kuramoto oscillators of varying coupling strengths which distribute onsets around periodic moments in time. We hypothesized that people tap regularly close to the sound onset density peaks when coupling is strong. When weaker coupling produces multiple inter-onset intervals that are more widely spread, people may interpret their variety and distributions differently in order to form a sense of beat. Experiment 1 with a small in-person cohort indeed showed a few individuals who responded with high frequency tapping to slightly weak coupled stimuli although the rest found regular beats. Experiment 2 with a larger on-line cohort revealed three groups based on characteristics of inter-tap-intervals analyzed by k-means clustering, namely a Regular group (about 1/3 of the final sample) with the most robust beat extraction, Fast group (1/6) who maintained frequent tapping except for the strongest coupling, and Hybrid group (1/2) who maintained beats except for the weakest coupling. Furthermore, the adaptation time course of tap interval variability was slowest in Regular group. We suggest that people's internal criterion for forming beats may involve different perceptual timescales where multiple stimulus intervals could be integrated or processed sequentially as is, and that the highly frequent tapping may reflect their approach in actively seeking synchronization. Our study provides the first documentation of the novel limits of sensorimotor synchronization and individual differences using coupled oscillator dynamics as a generative model of collective behavior.

**Data Availability Statement:** The data underlying the results presented in the study are available from https://ccrma.stanford.edu/∼nlem/swarm-tapping-study/

**Funding:** The author(s) received no specific funding for this work.

**Competing interests:** The authors have declared that no competing interests exist.

## Introduction

Our ability to sense temporal regularity within the soundscape and predict future events may have emerged as a survival mechanism, as natural environments contain an abundant source of sonic information in the form of auditory events [1]. Sensing regularity is also important when performing music in a group. In an ensemble context, group members construct a shared sense of time through *entrainment*, a process whereby we adapt and align our internal sense of timing with the external world. However, it is virtually impossible for multiple members to play notes strictly in synchrony even if their sense of time is perfectly aligned due to biomechanical and time-estimation errors [2, 3]. So, how do we interpret multiple sound onsets and construct a sense of pulse? What would people do if sounds are too widely spread and beat integration is no longer possible? In this study, we explored these questions through simple finger tapping with sound sequences that contain multiple sound onsets distributed around consecutive beat centers.

Human entrainment behavior has been extensively examined by using sensorimotor synchronization (SMS) tasks in which people typically tap in synchrony with regular auditory stimuli. The converging SMS findings include: people can reliably synchronize with a limited range of tempo (33–300 beat per minute (BPM), or inter-onset intervals (IOI) of 200–1800 ms) [4]; tap timings often precede stimulus onsets as the evidence of anticipatory behavior [5]; tapping is adaptive and dynamic at the beginning of the task or right after the stimulus perturbation such as phase shift or period shift [6–8]; experience enhances the accuracy of the tapping [9–11]. Interestingly, for the microtiming deviations typically observed in music, people appear to possess resilience and maintain a sense of regularity. For example, while listeners can detect a minute irregularity in an otherwise isochronous sequence, they still feel a sense of pulse in a sequence which contains twice as large deviations [12–14]. Altogether, we humans have abilities to feel regularity in sounds even if they are "quasi-periodic."

Research in speech perception and psycholinguistics have shed light on perception of quasi-periodic sounds in a similar timescale, employing a concept of "P-center" first proposed by Morton and colleagues [15]. P-center refers to the moment that a syllable is perceived to occur. Though never strictly isochronous, speech rhythms are perceived as somewhat isochronous by listeners across different languages [16]. In general, P-centers have been shown to correspond to a temporal region typically between the resulting acoustic waveform's perceptual onset and its acoustic energy peak for speech and music [17, 18]. Amongst recent P-center models related to music, the "beat bin hypothesis" [19] seems most relevant to our question about extracting regular beats from multiple streams of sound onsets. The beat bin model was originally conceived to explain how listeners maintain a sense of consistent beat in groove-based musical genres even though onsets are not perfectly quantized to a temporal grid. This hypothesis posits that "multiple onsets falling within the boundaries of the perceived beat will be heard as merging into one beat" resulting in the sense of the extended beat. Accordingly, as sound onsets are more sparsely distributed around a beat center, listeners' attention accommodates wider temporal intervals with diminished synchronization behavior [20]. Our question is how the structure of a sound's 'onset density'—the temporal spread of individual onsets around periodic moments in time—contributes to an expansion or contraction of synchronization behavior and how this influences beat extraction more broadly.

To characterize individual differences in synchronizing to sound onset densities, we employ a generative model of "coupled oscillators" [21] to generate auditory onsets that disperse around an average tempo. This dynamical system primarily describes a mechanical system of coupled oscillatory agents (like metronomes) that exchange energy to synchronize adaptively. Recently, similar oscillator models are proposed to explain adaptive coordination behaviors

between duet performers tapping together [22–24]. This could mean that individual agents may participate in the collective beat onsets by taking into account the coupling between their internal sense of the beat and the other sound inputs. In that sense, each performer in a musical ensemble has to couple with the others in the group, but their performance of the beat also influences the governing tempo. This coupling parameter in each individual member cannot be known without empirical investigation, and it may vary among individuals in population.

Individual differences in SMS, beat perception and production documented in the literature seem related to our present question. A wide range of abilities to tap in synchrony with music is related to the perceptual thresholds of timing deviation detection [25]. Further, strong and weak beat-perceivers are separated based on their responses to identical rhythmic patterns [26] where strong beat-perceivers showed increased brain activities measured by fMRI in part of rhythm-production related brain areas. Other individual differences in SMS may be attributed to musical training [27, 28]. Significant correlation between years of music training and beat-related behavioral performances [25] may be attributed to enhanced functional connectivities in musicians between the auditory cortex, supplementary motor area, premotor cortex, and putamen [29]. However, studies have shown that musical training is not the only determinant of individual differences. When examining only musically untrained participants [30], individuals who are good at maintaining a consistent spontaneous pulse still struggle to synchronize to external pacing stimuli, particularly for slower tempi (50 and 70 BPM). About 16% of young adults tested were classified as poor synchronizers with music but the majority of them performed adequately when tested on simple metronome and spontaneous regularity [31]. This proportion of poor synchronizers was almost identical in another study employing clapping and bouncing to musical excerpts with various beat saliency [32], while within the rest of normal synchronizers no correlation was found between synchronization regularity and music or dance training. Thus, the general population likely consists of people with a wide variety of perceptual beat tracking capabilities, regardless of musical experiences. However, naturalistic music excerpts or rhythmic stimulus patterns used in these studies were intended to examine how people perceive beats across multiple metrical levels including those with syncopated rhythms. What remains unclear is how people differ in integrating sounds that lack such hierarchical metrical relationships.

Therefore, in the present study, we asked people to tap with sounds with varied levels of coupling in the generative coupled oscillator model where they think the beat occurs. Fig 1 shows how the sound stimuli were produced from the generative model of coupled oscillators. In this 'swarm of points' representation, each oscillator is expressed as a point which rotates around a circle at an intrinsic frequency and increasing coupling strength forces the oscillators to phase align at different densities around periodic points in time. Sharp onset sounds (woodblock timbre samples) are produced upon each oscillator's cycle or zero crossing. Stronger coupling induces more tightly crowded oscillator swarms whereas weaker coupling distributes the oscillators more sparsely around the circle (in the 'None' coupling condition, oscillators simply move around the circle at their intrinsic frequency). We had three main hypotheses. The first was that participants would use the density of sound onsets in order to entrain their internal sense of beat. This predicts that stronger coupling conditions would result in more consistent tapping intervals and tap-stimulus phase relationships compared to weaker coupling conditions. In contrast, when the coupling is too weak, the sound onsets are widely distributed in time, making it difficult for listeners to hear any underlying temporal structure. The second hypothesis was that, as they discover the beat unfolding over time, their tapping behavior will show some adaptive characteristics, which would also interact with the coupling strength. The last and most important hypothesis was that, in discovering beats, individuals might employ different approaches, related to their internal tendencies that interact with

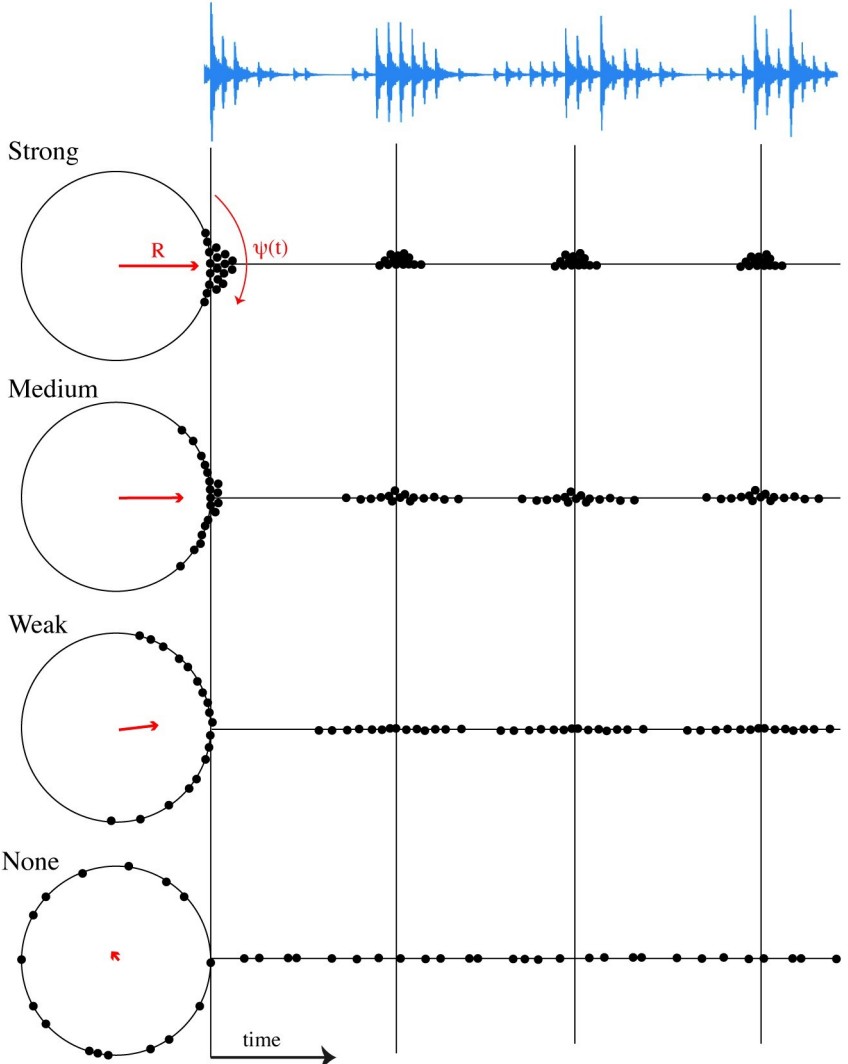

**Fig 1. Generative coupled oscillator model producing auditory stimuli.** Each oscillator in the generative model can be thought as a point moving around a circle and producing a sharp onset sound (e.g., woodblock sample) every time it completes one cycle. Stronger coupling produces onsets that are more densely crowded around periodic beat centers. When no coupling is applied, the oscillators simply move around the circle at their intrinsic frequency. The phase coherence, shown as a phasor with magnitude (R) and angle (ψ), provides an indication of the global synchrony of the oscillators in the group where larger values of R reflect tighter phase alignment.

external sounds of different coupling strength. When confronted with multiple inter-onset intervals, people may interpret their variety and distributions differently in order to extract a sense of beat.

Two experiments were conducted. Experiment 1 served as an in-person, small cohort pilot experiment to investigate first whether our paradigm including stimulus generation, apparatus and instruction, and analysis methods could capture people's naturalistic tapping behavior. This experiment involved three stimuli coupling conditions (strong, medium, and weak). Experiment 2 extended the stimulus set by allowing for another coupling condition ('None', or no coupling). It also allowed for increased tempo variation and stimulus sequence duration. Moreover, we recruited a larger cohort online but using the same task platform as Experiment 1 to analyze different tapping patterns in greater detail.

## Experiment 1

This experiment served as a preliminary proof of concept as we wished to determine whether participants could perform tapping with a simple keypress with a laptop computer. Also, we wished to test whether participants could perform the task under minimal instruction such as 'tap where you feel the beat while listening to sounds' in order to allow people and document their wide variety of naturalistic behavior rather than giving potentially constraining or suggestive instructions. As such, we conducted this experiment as an in-person study for a small-cohort of participants within our local community at the university.

## Methods

### Participants

10 volunteers from the Stanford University community took part in this study (5 Women, $M_{age}$ = 31 years, $SD$ = 5.6 years, all right handed). 9 of them had musical training ($M$ = 10.1 years, $SD$ = 4.2 years). Participants were requested to review and sign an informed consent document and were compensated with gift cards for their participation. Stanford's Institutional Review Board approved our experimental procedure.

### Stimuli

Audio stimuli were generated from a network of phase-coupled Kuramoto oscillators [21]. Below, we first provide a short overview of the dynamics associated with the system. Thereafter, we will describe the audio synthesis methods using the model output to generate the auditory stimuli.

Coupled oscillators are a type of dynamical system that models the behavior of typically large-scale interactive and synchronizing systems. They encompass a wide range of self-organizing populations that have been observed in biological, chemical, and electrical systems [33]. They have been used to describe the synchronous behaviors characterizing cortical oscillation as well as providing a model to describe the oscillatory neurodynamics of rhythmic entrainment in music [34]. A basic mathematical model that describes coupled oscillation is known as the Kuramoto Model which concerns an ensemble of limit-cycle oscillators interacting at the phase level. The governing equation is shown in Eq 1.

$$\dot{\phi}_i = \omega_i + \frac{k}{N} \sum_{j=1}^{N} sin\left(\phi_j - \phi_i\right), i = 1 \ldots N$$

where $\phi_i$ is the phase of the $i^{th}$ oscillator and $\dot{\phi}_i$ the derivative of phase with respect to time. $\omega_I$ is the intrinsic frequency of that particular oscillator, among a population of $N$ oscillators. $k$ is the coupling coefficient and the $sin(\phi_j - \phi_i)$ term is known as the phase response function. This non-linear function determines the interaction between each oscillator and the ensemble as the network forms a complete graph topology (i.e., each connected with each one of the others). When the ensemble's intrinsic frequencies follow a Gaussian distribution and when a critical value of coupling is applied, the oscillators' phases begin to align and synchronize to a frequency near the center of the initial intrinsic frequency distribution. This state transition into periodic synchrony can be illustrated by evaluating the complex order parameters—the phase coherence, $R$ and average angle, $\varphi$—two summary statistics that are convenient for observing the global behavior of the ensemble at large. These two parameters are related to the

oscillators as below in Eq 2.

$$Re^{i\varphi} = \frac{1}{N}\sum_{j=1}^{N} e^{i\phi_j}$$

By modifying the ensemble's coupling, we can force the system into different quasi-periodic states that reflect the synchrony of the ensemble. It is useful to imagine the oscillators in the ensemble as a "swarm" or "crowd" of points rotating around a circle. In this circle map paradigm, full synchrony is achieved when the swarm of points is moving around the circle at the same rate and the same instantaneous phase angle (in terms of the complex order parameters, this implies that $R = 1$ and the derivative of the average angle, $\dot{\varphi}$, with respect to time is constant. It should be noted that in Eq 2, $i$ refers to a complex number). We can then use the complex order parameters to rewrite the governing equation (Eq 1) as a function of $R$ and $\varphi$ as shown in Eq 3 below.

$$\dot{\phi}_i = \omega_i - kR\,sin(\varphi - \phi_i)$$

In order to apply this model to sound, we allow each oscillator in the ensemble to trigger a prepared audio sample at each zero crossing of its instantaneous (wrapped) phase (when $\phi_{i,t-1} < \phi_{i,t}$ where $t$ denotes the current iteration).

The oscillators' individual zero crossing timings were computed and saved to separate files in order to 1) reconstruct the complex value of the phase coherence angle ($\varphi$) of these oscillators that we later use to define the beat windows per stimulus sequence, and 2) compare the participant's tap response and the stimulus' oscillator zero crossings within beat windows for phase coherence analysis (for details, see S1 Fig and the corresponding section in the S1 File). The specific implementation of the generative model was written in the Python programming language and used a forward Euler numerical integration scheme at a temporal precision of $\Delta t = 0.016$ s. We approximated the sound of a metronome by using an audio sample of a struck "woodblock." Without any coupling applied, each oscillator produces a simple "metronome" sound once per cycle as determined from its intrinsic frequency. With an increased coupling parameter, more oscillators with intrinsic frequencies near the center frequency of the initial distribution are recruited into entrainment.

Experiment 1 used an ensemble of forty coupled oscillators with three coupling conditions (strong, medium, weak) and five tempo conditions (90, 95, 100, 105, 110 BPM). Here, these coupling coefficients were kept constant with respect to time throughout the course of the audio generation, resulting in a slight fluctuation in phase coherence throughout each sequence (To stabilize such fluctuations, in Experiment 2 the coupling coefficient was allowed to be time-variant via feedback). The way in which the generative model produces onset timepoints and how those onsets relate to the stimuli is shown in examples in Fig 1. Out of numerous output sequences, we selected the final set of sequences based on their average phase coherence magnitude ($R$) ranges; strong (0.79 - 0.85), medium (0.55 - 0.57), and weak (0.14 - 0.27) coupling conditions. In order to discourage the participants from memorizing the audio stimulus for each condition, we created two auditory stimulus versions for each combination between tempo and coupling condition. Depending on the condition, each auditory stimulus contained nine to fourteen beats. This resulted in a total of 30 audio stimulus files.

## Test procedure

In a sound proof studio, participants were asked to tap on the spacebar of a 12" MacBook Pro Laptop using the dominant finger of their dominant hand. The experiment was created using

PsychoPy (v.3) [35] software which allowed presenting the audio files and recording of each tap timing as a space bar press per stimulus sequence. Upon beginning the experiment, the participants encountered the following experimental prompt on the screen: "You will hear a sequence of tones. Use the index finger of your dominant hand to tap the spacebar at where you think the beat is." The auditory stimuli were presented diotically (i.e., same sound to both ears) to the participants using headphones (Sony MDR-7506). In order to allow the participants to become familiar with the experimental setting, they first performed a practice trial block consisting of five auditory stimulus sequences. The 30 audio files for the actual testing block were then presented to participants in a randomized order for a total duration of around 12 minutes. The same block, with a different sequence order, was repeated after a short break.

It is important to note the timing precision capacity of our apparatus. For PsychoPy, a recent comparative analysis [36] documented a less than one millisecond precision for macOS in reaction time (0.4 ms) and audio onset (0.7 ms), although the latency is 22ms and 0.5ms respectively. The latency does not affect our main questions about the interval between subsequent tapping and the tapping coherence vs. stimulus onset coherence through a trial. However, this slow latency did not allow us to compute the signed timing difference between each one tap and one stimulus onset (i.e., mean asynchrony). The inherent latencies of the computer keyboard to register taps (8ms which is the inverse of the polling rate of 125Hz) do not affect our analysis. Thus the high precision of PsychoPy reaction time registration on the laptop sufficiently supports our data collection and analysis.

## Data analysis

The generative coupled oscillator model produced stimuli by using different coupling strengths to distribute sound onsets around periodic moments in time. Briefly, the generated stimuli's average angle, $\varphi$ at each time step, was used to determine beat window by our segmentation algorithm extracted the referent beat centers. The tapping data for each stimulus sequence were extracted as inter-tap interval (ITI) for each tempo and coupling condition, and circularly mapped onto the beat windows in order to perform phase coherence analysis and compare with the aforementioned stimuli's phase coherence. Below, we detail each of these analysis procedures.

The participant taps were analyzed in two ways: ITI analysis and phase coherence analysis. ITI refers to the amount of time (period) between two successive taps per beat section. We divide these intervals by the mean period of the stimuli to derive a normalized ITI (nITI) to handle the data across different tempo conditions. This means that if a person taps in perfect alignment with beat windows in a given stimulus sequence, their average nITI in that trial would be 1. This normalization was necessary to integrate data across different tempo conditions and to capture beat-based tapping behaviors. As nITI histograms for the strong, medium and weak coupling conditions show bimodal distributions, we further determined the individuals' histograms. This revealed two participant categories, which we called the Regular tapping group ($N = 8$) and Fast tapping group ($N = 2$). The former group's histogram data across coupling conditions centered around an nITI of 1, while the latter groups centered around 0.2. In order to analyze the evolution of their tap performance over time, we then divided the first nine nITIs into three tap sections (each one accounting for three tap intervals) and calculate the mean and SD of the nITIs, to show how the tap intervals and variability change over time within a given trial.

The phase coherence analysis was performed for participants' taps via complex order parameters (phase coherence and average angle) per beat window and compared to the complex order parameters of the stimulus, an ensemble from the generative model. This allowed

us to analyze how the participants' taps lined up with the center of the beat as defined by the beat window generated from the beat window segmentation algorithm.

For each auditory stimulus from the generative model, we treat each oscillator zero crossing as a "stimulus onset" and then translate these timepoints to a circle map representation defined by the stimulus beat windows. This provides us with a phase angle that denotes where the oscillators zero crossings are situated within each beat region. Finally, we use Eq 2 to determine the $R$ and $\varphi$ for each coupling condition; this can be depicted as a phasor with magnitude $|R|$ and angle $\varphi$. The same analysis is done for the participants' taps and the resultant complex order phasors are compared between the participant taps and stimulus onsets.

To examine all tap and onset distributions across one beat cycle required circular statistics. We used Rayleigh's test, to assess for the uniformity of the distributions under inspection [37]. This test statistic is useful as it discourages making a priori assumptions about a unimodal departure or a pre-specified mean direction.

## Results

### Normalized inter-tap-intervals and participant differences

All participants successfully produced multiple tapping responses for each sequence. As shown in Fig 2A, the nITI histogram from 10 participants show clear bimodal distributions in all three coupling conditions. Two participants tapped around four times per beat window (nITI ≈ 0.25) across all coupling conditions (darker color histograms in Fig 2A). We call these two participants the Fast tapping group, and the remaining 8 participants the Regular tapping group (as their nITIs were close to 1). Interestingly, histograms show that the distribution of the taps in the Fast group becomes narrower as the coupling weakens whereas the Regular group's distribution was narrowest for the strong coupling, becoming wider for weaker coupling.

### Tapping adaptation and phase coherence

Fig 2B shows the time course for the mean nITI and SD nITI across three tap intervals over each of the three tap sections within a given sequence, separately for the two groups. For the Regular group, their mean nITI tracked closely with the stimulus in the medium and strong conditions and their tap variability across three intervals tended to decrease over the course of the stimulus duration. The tap interval variability in the weak condition was also reduced over the tap sections, but its slope appears shallower. For the Fast tapping group, tap responses largely hovered around nITI of 0.3 regardless of coupling. Tap-to-tap variability over time decreased similarly for the strong and medium coupled conditions, but the variability was elevated for the strong and medium conditions, which contrasts with the tendency in the Regular group.

We examined these time courses only for the Regular tapping group using a two-way repeated measures ANOVA in three different tap Sections (early, middle, and late) and three Coupling strengths (strong, medium, weak). For the mean nITI, the ANOVA revealed no significant main effects or interaction. The main effect of Section was marginal ($F(2, 14) = 2.73$, $p = 0.099$, $\eta_G^2 = 0.105$, N. S.), as was the two-way interaction ($F(2, 14) = 2.59$, $p = 0.058$, $\eta_G^2 = 0.099$, N. S.). The same design ANOVA for the SD again revealed no significant effects.

Fig 2C illustrates the phase coherence of the participant taps with the phase coherence of the stimulus onsets. For the Regular tapping group, comparing the phase coherence for each coupling condition to that of the stimuli showed that the length of $R$ increased from weak to medium and strong coupling, while the angle, $\varphi$, decreases with increasing coupling. For the Fast tapping participants, we observe more variance of tap placement within the circular beat window. All of the stimulus onsets contained significant directionality ($p < 0.0001$) by

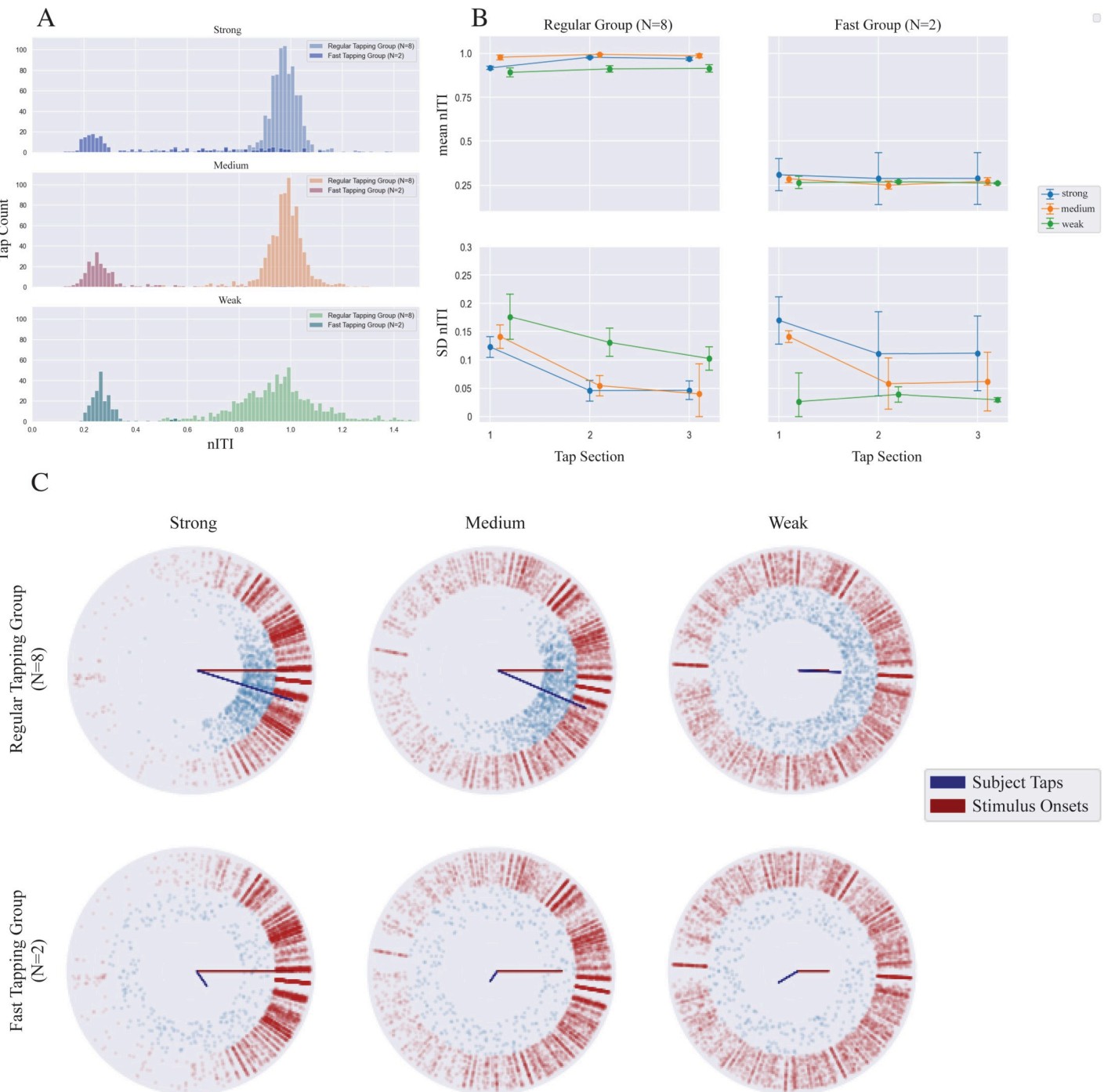

**Fig 2. Tapping data in Experiment 1.** A: Histogram of normalized ITI (nITI) for three coupling conditions (strong, medium, weak) for the Regular tapping group (N = 8, light colored bars) and the Fast tapping group (N = 2, dark colored bars). B: Changes in tapping behavior over the course of three tap sections, each contain 3 tap intervals from the beginning of a trial. The mean and SD of nITI data (top, and bottom, respectively) for three coupling conditions (strong, medium, weak), in the Regular tapping group (left panel) and Fast group (right panel). The error bar represents standard error of the mean (SEM) within group. C: Phase coherence (R) and average angle (φ) of participant taps (blue) and stimulus onsets (red). These circular plots have been shifted so that the stimulus φ is centered at 0 degree. Each blue dot corresponds with a participant's placement of their tap within each beat window for each stimulus; each red dot corresponds to a stimulus sound-event onset that is a result of a single (coupled) oscillator from the generative model. Similarly, the red and blue vectors are the phase coherence magnitudes (R), a summary statistic that indicates the level of synchrony for the stimulus onsets and participants' taps as a whole over the duration of each stimulus. In this orientation, φ < 0 implies tap leading; φ > 0, tap lagging.

Rayleigh tests. For the tapping data, the Regular group's taps showed significant directionality ($ps < 0.0001$) for all three coupling conditions, whereas for the Fast group, only the strong coupled condition resulted in significant directionality ($p < 0.0001$) but not in the medium and weak coupled conditions.

## Discussion

We found that beat extraction behavior differed between two individuals with highly frequent tapping and the remaining eight participants who largely maintained steady beats close to the oscillators' base tempo. On the one hand, this observation supported our idea of observing inter-individual differences in extracting beats from concurrent, multiple onsets, with open-ended instruction. On the other hand, because of the small cohort, it is unclear whether these two individuals were outliers specific to our sampling or a legitimate category of individuals as part of the general population. Engaging a larger cohort and a wider range of coupling conditions might help distinguish these cases while also investigating their characteristics with more statistical power.

We did not observe tangible adaptive behaviors when statistically examined in the Regular group, likely due to three factors. First, we had only a small subgroup. Second, our stimulus sequences might have been too short for observing beat discovery processes. Third, our tempo range of 90–110 BPM might have been too narrow, possibly allowing participants tap from memory, or use their own spontaneous tapping rate, which previous literature has shown to be close to our chosen tempo range [38, 39]. In addition, it could also be fruitful to examine how participants responded to stimuli that contained no coupling as a baseline condition. Therefore, Experiment 2 was designed to investigate these possibilities more clearly and robustly with improvements in the design.

## Experiment 2

This second experiment was aimed at improving the task design for a larger cohort of participants. Specifically, for the task design, we extended the tempo variations across trials so that participants could not rely on memory or spontaneous tapping rate. Also, we extended the stimulus sequence duration to observe adaptive processes over the course of the beat discovery more thoroughly. Lastly, we extended the stimulus conditions to include the 'none' coupling condition, where the intrinsic frequency of the oscillators are at the given base rate, and the onset density peak is nonexistent. This condition was included in order to examine how the associated tapping patterns are different from the three weak to strong coupling conditions. We kept the same instruction and computerized task setup as those in Experiment 1 while making the experiment accessible online via a web browser. The online environment was implemented for two reasons. Firstly, the pandemic-related restriction made in-person human participant research in our environment impossible. Secondly, we wished to include a larger group of participants outside of our local community. Although we were aware of the variety of participants' computer systems, we are confident that our data collection and analysis methods are valid given the high precision across different OS systems for our chosen software platform [36] and given the reliability of the inter-tap intervals in our analysis. Practice trials were employed before the actual trials for the dual purpose of introducing the task to the participants and screening individuals' tap compliance.

## Methods

### Participants

81 participants enrolled in this online experiment. They were recruited via online advertisements circulated in both the Stanford University community and Mechanical Turk (MTurk). After screening based on the performance of the practice trials, 50 participants made up the final sample. Specifically, our screening criteria were that they had to tap more than 80% of the assumed number of taps in the strong coupled practice trials and multiple taps should be made for all other practice trials. The screening was necessary because the online task was made such that one could proceed through all the trials without pressing any key during the stimulus playback, and the participants took the test without an experimenter's real-time supervision. This final sample size exceeds the estimate of 30 for discovering a significant $F$-statistic in a mixed ANOVA between-within factor interaction assuming 2 between-subject groups and 4 coupling levels of a within-subject factor (Cohen's $f$ = 0.25, $\alpha$ = 0.05, Power = 0.9), as estimated using G-power (ver 3.1.9.7) [40]. Demographic questions before the block of tapping trials were also not mandatory to proceed and complete the task, and some participants left answers blank. Our final sample's demographic information is as follows: Sex (all 50 answered the question): 32 men, 18 women; Age (all answered the question): $M$ = 35.7, $SD$ = 11.2, Range: 20–66; Years of musical training (self-interpretative, 41 answered the question): $M$ = 6.1, $SD$ = 8.7, Range: 0–30. Participants recruited in the local community were invited to review and download a digital information sheet about the study before proceeding to the experimental webpage and were compensated with electronic gift cards for their participation. Participants from MTurk were shown the same information as part of the task advertisement at MTurk site and compensated with cash in their MTurk account. Stanford's Institutional Review Board approved our experimental procedure.

### Stimuli

Audio stimuli were generated using the same generative model of forty phase-coupled oscillators in Experiment 1. The oscillators' individual zero crossings were also saved in separate files in order to reconstruct the complex order parameters that are used to define the beat windows. For this experiment, we generated stimuli representative of four different coupling conditions—strong, medium, weak, and none (in which the oscillators were not coupled with one another)—and five different tempo conditions (72, 81, 92, 105, 119 BPM) to create a total of 40 audio stimuli per block. Five additional sequences for practice trials were also created (2 strong, 1 medium, 1 weak, and 1 none coupling condition at 81, 72, 105, 92, and 119 BPM respectively) to familiarize participants with the apparatus and task, before the test trials. The range of the average phase coherence $|R|$ in these sequences are: strong (0.95 - 0.98), medium (0.34 - 0.57), weak (0.27 - 0.42), and none (0.05 - 0.18). This time, the generative model's behavior was controlled by a feedback system to stabilize the fluctuation of phase coherence, making the global couple coefficient, $k$, either incremented or decremented by a constant value to step the phase coherence toward the target value. As in Experiment 1, the computed oscillators' sound onset timing data stored in separate files were analyzed to extract beat windows. Since any synchrony that arises from the none-coupling condition is local and random, and the trajectory of $\varphi$ over time is less well defined, we imposed the algorithmic constraint that the beat window segmentation should only count zero crossings that are less than twice the mean period of all the oscillators in the ensemble. Thus, the resulting beat-window sizes were close to the averages of instantaneous oscillator periods. Again, this does not affect our nITI analysis because the position of the beat window becomes arbitrary. We increased the

duration of each sequence so that it corresponds to 22 beats long with respect to the oscillators' base tempo.

We created two different sets of 40 auditory stimuli such that one set contained these stimulus sequences and the other contained 40 sequences with small timbre differences between the oscillators' sounds. The latter set is designed for a different study, and the corresponding data will not be analyzed in this paper. These two sets were presented separately in two blocks but in a counter-balanced order across participants. Participants were not informed about the nature of the difference between the two blocks, except that they were instructed to have a short break between them. The total duration of the task was about 30 minutes.

## Test procedure

The experiment was conducted online via a web browser using PsychoPy JS [41], a newer version of PsychoPy which was used in Experiment 1. PsychoPy JS is capable of providing an online version using Javascript that works on a web browser. The advantage of this is that when participants are performing the task, it downloads the relevant files temporarily to participants' local computer to operate stimulus presentation, key press monitoring, process control, and working data maintenance. This provides us with a comparable environment across participants between Experiments 1 and 2, despite admittedly having no control over participants' computer setup. Given the consistently high precision across different computer OSs, our assumption is that timing characteristics in an individual's setup would have affected the data from the same individual across all conditions, rather than some of the conditions. When participants open the task URL, they were first asked to input demographic information into a dialog pop-up box, then they proceeded to the practice trials and experimental trials. In each of these trials, a stimulus sequence (presented as an audio file in mp3 format) is played back diotically (i.e., same for both ears), and the participants were asked to tap on the spacebar of their personal computer using the dominant finger of their dominant hand. The participants were given the same experimental instruction as in Experiment 1 and were requested to use headphones during the study. After the five practice trial sequences to familiarize the participants with the experimental task, two blocks of each forty experimental stimuli were presented in a randomized order.

## Data analysis

We used the same analysis techniques from Experiment 1 to extract beat windows for each experimental stimulus sequence. Next, we examined distributions of produced nITIs to see whether the tapping behavior shows distinct patterns between individuals. Specifically, nITIs averaged within each sequence per participant were plotted as function of the stimulus coupling strength expressed with stimulus phase coherence magnitude. This 2D map of data points were then subjected to $k$-means clustering ($k$ = 5) (Fig 3).

For the clustering, the number of clusters ($k$ = 5) was chosen because it separates the trials into two clusters with a nITI around 1 (one with the strong coupling condition and the other weaker ones), one cluster that accounts for high frequency tap trials, and two clusters that account for low frequency tap trials. The first three clusters contained more than 98% of all cases. Specifically, clusters 1 and 2 represent nITIs close to 1.0 in the strong and weaker coupling regions respectively, while cluster 3 occurs around the weak to none coupling around nITIs shorter than 1.0. Thereafter, we applied the following criteria to characterize categories of individuals: we classed individuals as belonging to the Regular tapping group if more than 50% of their tapping data belonged to clusters 1 and 2; the Fast tapping group if more than 50% of the time their tapping belonged to the cluster 3 except for the strong coupling

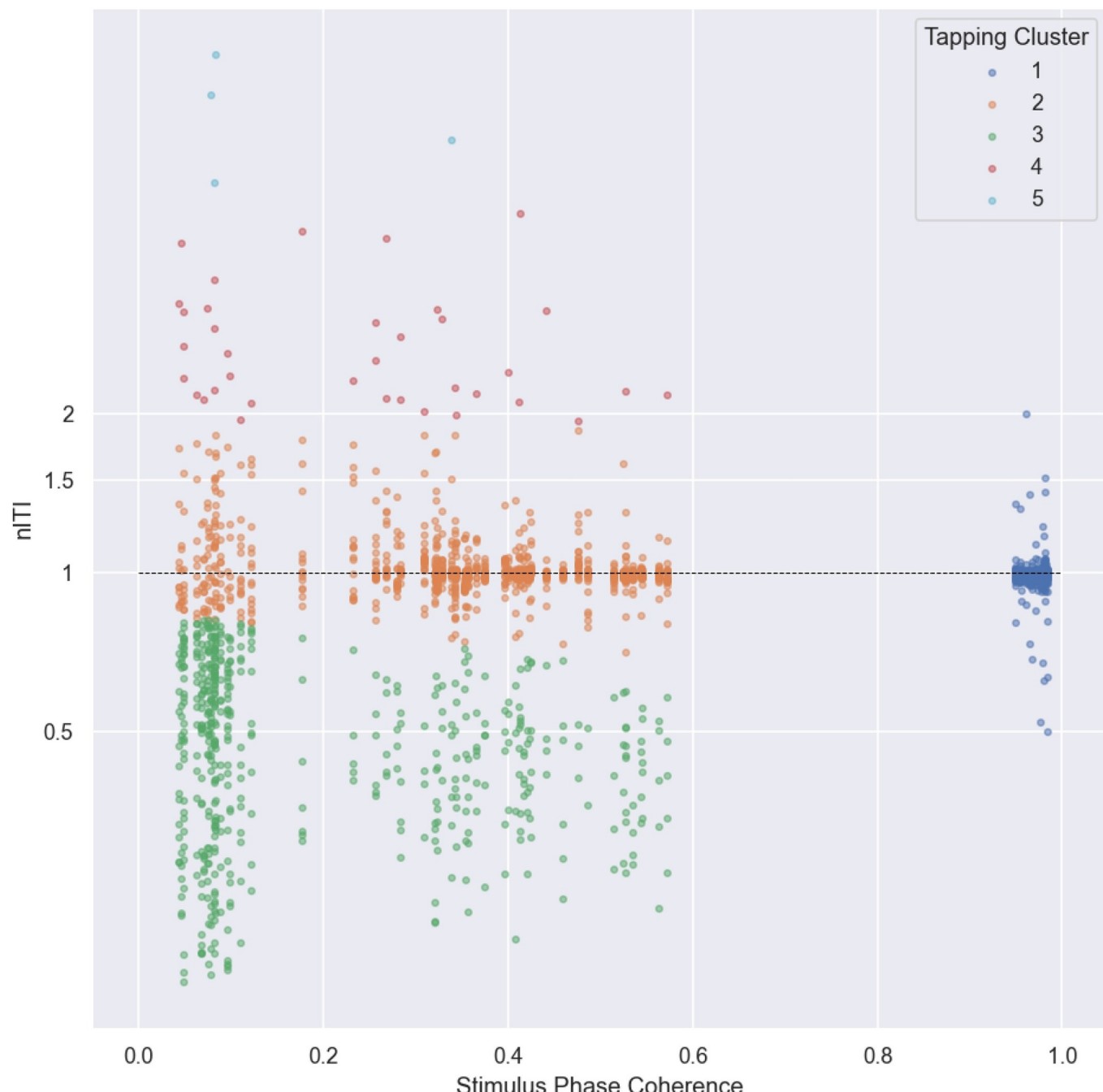

**Fig 3. Results of k-means clustering of tapping data as function of phase coherence magnitude of stimuli.** Each point reflects a participant's mean nITI over each stimulus sequence. 98% of the data points fell into tapping clusters 1, 2 and 3.

condition which makes up the cluster 1; lastly, the Hybrid group if their tapping data showed more points in the cluster 3 only at the none-coupling condition (Table 1).

The nITI distribution for different coupling conditions were then visualized in histograms separately for these three groups (Fig 4A). The mean and variability of nITIs were examined to see the adaptive patterns of the tapping over the course of the stimulus sequence, separately for these three groups. This time, each group's performance was assessed with repeated measures

**Table 1.  The percentage of taps (mean (SD)) in different clusters in three tapping pattern groups.**

|  | Coupling | Cluster 1 + 2 | Cluster 3 | Cluster 4 | Cluster 5 |
|---|---|---|---|---|---|
| Regular Tapping group (N = 17) |  |  |  |  |  |
|  | Strong | 100.0 (0.0) | 0.0 (0.0) | 0.0 (0.0) | 0.0 (0.0) |
|  | Medium | 95.6 (11.0) | 0.0 (0.0) | 4.4 (10.6) | 0.0 (0.0) |
|  | Weak | 91.0 (17.5) | 19.0 (17.1) | 7.5 (17.1) | 0.0 (0.0) |
|  | None | 59.4 (19.5) | 29.4 (24.5) | 10.0 (24.5) | 1.25 (3.3) |
| Hybrid Tapping group (N = 24) |  |  |  |  |  |
|  | Strong | 99.6 (2.0) | 0.0 (0.0) | 0.0 (0.0) | 0.0 (0.0) |
|  | Medium | 93.6 (11.6) | 6.0 (12.0) | 0.0 (0.0) | 0.0 (2.0) |
|  | Weak | 94.4 (11.3) | 4.8 (11.0) | 0.0 (2.0) | 0.0 (0.0) |
|  | None | 21.2 (15.0) | 78.0 (15.0) | 0.0 (2.0) | 0.0 (2.0) |
| Fast Tapping group (N = 9) |  |  |  |  |  |
|  | Strong | 100.0 (0.0) | 0.0 (0.0) | 0.0 (0.0) | 0.0 (0.0) |
|  | Medium | 14.4 (17.7) | 85.6 (17.7) | 0.0 (0.0) | 0.0 (0.0) |
|  | Weak | 5.6 (9.6) | 94.4 (9.6) | 0.0 (0.0) | 0.0 (0.0) |
|  | None | 1.1 (3.1) | 98.9 (3.1) | 0.0 (0.0) | 0.0 (0.0) |

ANOVAs using two within-participant factors of tap Section (1–7, each containing three consecutive ITIs) and Coupling (strong, medium, weak, none). Further, phase coherence analysis was performed for taps and stimulus onsets. Rayleigh tests were used to assess the uniformity of a circular distribution of each group's tap data in each condition, as well as the stimulus onsets in each condition. Watson-Wheeler tests were used to compare participants' taps and stimulus onset circular distributions.

For cluster 3, we performed an additional analysis to see if their high frequency tapping responses reflect the effects of the base tempo and stimulus coupling separately. While stimulus tempos are reflected in beat-based tapping intervals when a regular beat is extracted, we wanted to determine whether their frequent tapping performance was still sensitive to the stimulus tempo, coupling, or both. Importantly, when the stimulus coupling is decreased, the distribution of onsets in the stimulus would naturally increase temporal spread, which causes in turn changes in stimulus IOIs in addition to the effect of the base tempo. To answer this question, we used the consecutive raw (i.e., un-normalized) ITIs and examined a two-dimensional 'phase portrait' representation related to stimulus beat centers (see S2 Fig and the corresponding section in the S1 File). Then, the cluster 3 tap data were further categorized into 'dense' tapping patterns in which consecutive tap intervals did not vary much, and the 'sparse' tapping patterns for the remaining occurrences in each individual participant. These categories were based on the split point of the dispersion metric's bimodal distribution, which was computed from the phase portrait representation. Finally, using multiple regressions, we determined whether the dense and sparse tapping ITI data exhibited the influences of base tempo, participant grouping, and stimulus coupling.

## Results

### Normalized inter-tapping-interval clustering and participant categories

Participants' average nITIs were shown in Fig 3, plotted as a logarithmic function of the phase coherence of the stimuli. *K*-means clustering revealed five different regions: cluster 1 indicates the strong coupling condition and nITI around 1.0, reflecting the regular beat tapping. Cluster 2 shows the same regular beat tapping but for the weaker coupling conditions. Cluster 3 shows

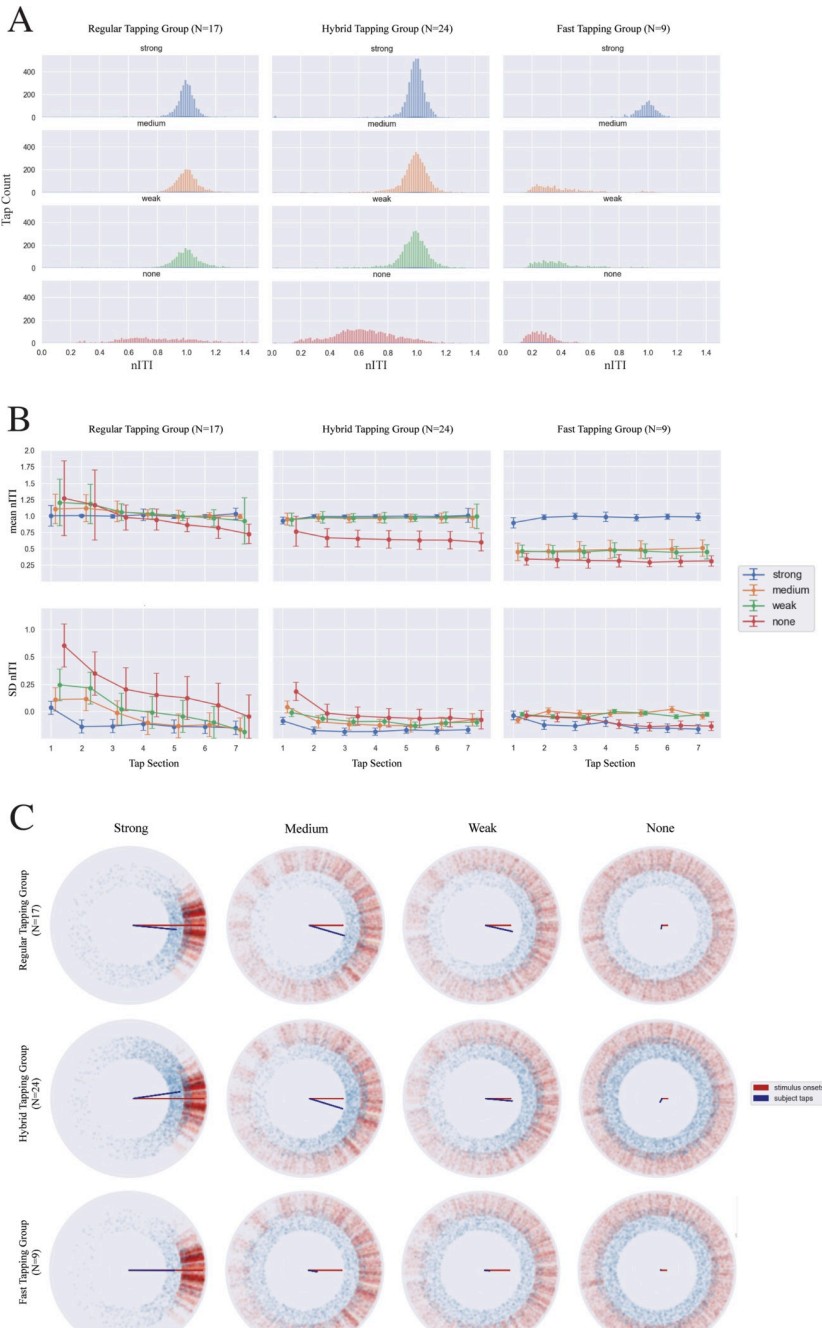

**Fig 4. Tapping data in Experiment 2.** A: Histogram of the participants' average nITIs for the three tapping groups (column) for four coupling conditions (row). All of the tapping groups tapped at nITI centered around 1.0 for the strong coupled stimuli. Regular and Hybrid groups tended maintain this tapping approach when presented with medium, weak coupled stimuli, except for Hybrid switching to much shorter nITI with the none-coupled stimuli. Fast group began to tap with shorter nITIs with any of the weaker coupled stimuli (medium, weak, and none). B: Tapping adaptation in mean nITI (top) and standard deviation (SD) (bottom) over the course of stimulus sequence for Regular, Hybrid, and Fast groups. The values were calculated using the first 21 tap intervals, separated in 7 sections each containing three tap intervals, from the beginning of the stimulus sequence. The error bars express the standard error of the mean (SEM) within group. C: Phase coherence (R) and average angle (φ) of participant taps (blue) and stimulus onsets (red). These circular plots have been shifted so that the stimulus φ is centered at 0 degree. Each blue dot corresponds with a participant's placement of their tap within each beat window for each stimulus; each red dot corresponds to a stimulus' sound-event onset that is a result of a single (coupled) oscillator from the generative model. Similarly, the red and blue vectors are the phase coherence magnitudes (R), a summary statistic that indicates the level of synchrony for the stimuli onsets and participants' taps as a whole over the duration of each stimulus.

the parallel distribution to cluster 2 for the weaker coupling conditions but reflecting frequent tapping with nITI less than 0.7. Clusters 4 and 5 show rare cases for nITI of around and more than 2, respectively. Notably, 98% of all tap responses fell within the first three clusters.

The three groups of individuals were identified based on the criteria of how their tapping strategy changed across different coupling conditions. The Regular tapping group ($N = 17$) making up about a third of the final sample of 50, consisted of individuals whose data belonged 50% or more to the clusters 1 and 2. In contrast, if individuals switched into a faster tapping behavior in cluster 3 at the medium coupling condition, and continued to keep the frequent tapping strategy for the weaker conditions, this group is termed as the Fast tapping group ($N = 9$), about a sixth of the final sample. The rest consisted of those who maintained regular tapping strategies except for switching to frequent tapping cluster 3 more than 50% of the time in the none coupling condition. We termed this group as the Hybrid group including approximately a half of the final sample ($N = 24$).

Table 1 showed group means and SDs for the three groups' tapping distributions across the clusters. For the Regular tapping group, over 90% of their tap responses for the three coupled conditions belong to clusters 1 and 2, but that was reduced to 59% for the none coupled condition, with the increased response in the cluster 3 to 29%. For the Hybrid group, over 90% of their taps were sorted into clusters 1 and 2 for stimuli in the three coupled conditions, whereas around 78% of their taps to the none-coupling stimuli were sorted into the cluster 3. Notably, 21% of their taps in the none-coupling condition on average remained in the regular clusters 1 and 2, showing that this group remained slightly affiliated with the regular beat processing. In the Fast tapping group, the majority of tap responses for the three weaker coupling conditions belongs to the frequent tapping cluster, increasing from 85% for the medium, to 94% for the weak, to 99% for the none-coupling condition. Note that these proportion data per group are made for a descriptive purpose but not for inferential statistical testing, because the same data in the clustering procedure provided the criteria for classifying participants into groups.

These observations for the three groups are associated with the differences in histograms of each group's nITI, as shown in Fig 4A. In general, all participants tapped at a nITI of 1 for the strong coupled stimuli. For the three coupled stimuli (strong, medium, weak), the Regular group's distribution was narrower relative to the other tapping groups, and their distribution widened for decreased coupling. In contrast, in the none-coupled stimuli, the Fast tapping group had the narrowest tapping distribution. With respect to the Regular and Hybrid groups, histogram shapes for the strong, medium, and weak conditions are comparable.

We examined whether any demographic differences exist between three groups. For Regular, Hybrid and Fast groups respectively, the mean age (*SD*) was 37.3 (12.6), 35.2 (10.9), and 34.1 (10.1). Age data showed no significant difference among groups, compared by a Kruskal-Wallis test ($H(2) = 0.39$, $p = 0.82$). The gender makeup of the tapping groups (all 50 participants answered the question) are as follows (F/M): Regular (6/11), Hybrid (9/15), and Fast (3/6). Self-report handedness is as follows (R/L): Regular (15/2), Hybrid (16/8), and Fast (8/1). Chi-square test revealed no association between group affiliation and sex ($\chi^2 = 0.059$, $p = 0.97$) or handedness ($\chi^2 = 3.46$, $p = 0.18$). Within the participants who reported their years of musical training within the Regular group, there was a large disparity between those having none or very little musical training (10 people with 0–3 years) and extensive training (5 people with 18–30 years, 2 did not report this information). Among those who reported their musical training in the Hybrid group, 10 people had 0–2 years of musical training, 3 had 6–8 years of musical training, and 1 had 30 years of musical training. Among those who reported their musical training in the Fast tapping group, all seven reported having 0–3 years of musical training. A Kruskal-Wallis test showed that the group differences in musical training were not significant ($H(2) = 2.41$, $p = 0.30$).

## Tapping adaptation and phase coherence

In order to capture how participant nITIs evolve over time, we placed the participants' first twenty-one tap intervals into one of the seven tap sections. The mean nITI and SD for each section are shown in Fig 4B separately for three groups. Notably, the adaptation trajectories, in particular for the SD, are different between three groups; tap variability was highest at the beginning of the sequence in the Regular group which is reduced over time, whereas the curves were shallower in the Hybrid group, and the variability change remained flat or slightly upwards in the Fast group.

Fig 5A shows the exponential curve fitting to examine the group differences for the SD of nITI over tap section, while Table 2 shows details of the estimate and significance with the $R^2$ goodness of fit. The $A$ (intercept) and $B$ (decay rate) coefficients for the exponential, $Ae^{B*t}$, are shown in Fig 5B. We plotted them against the magnitude of the stimulus phase coherence for each tapping group. As seen in Fig 5B, Regular and Hybrid groups differed in both intercept and decay rate changes over the coupling conditions. The Regular tapping group starts off with the most tap-to-tap variability as reflected by their larger intercept value when the stimulus phase coherence magnitude is low. At the same time, the intercept falls off sharply over increasing stimulus phase coherence magnitude showing how they reduce their tap variability over time more quickly than the two groups. Conversely, the decay rates were most negative for the Regular tapping group because their curves dropped off with the fastest rate of change. However, in the Fast tapping group, the medium and weak tap-to-tap variability increased from the beginning tap section to the end which is reflected in their decay rates being positive for $|R_{stim}|$ in the medium to weak coupling condition range (e.g., 0.4–0.6).

We used repeated measures ANOVA with Coupling (strong, medium, weak, none) and Section (1–7) separately for each group to characterize the mean and SD nITI change over the course of the sequence. For Regular group's mean ITI data, the main effect of Coupling was significant ($F(3, 48) = 2.86$, $p = 0.047$, $\eta_G^2 = 0.035$) and the post-hoc tests showed that the nITI was significantly longer for the strong coupling condition compared to the weaker coupling conditions (for details of post-hoc test results, see S1 Table). Also, the main effect of Section was significant ($F(6, 96) = 6.79$, $p < 0.001$, $\eta_G^2 = 0.13$) as the earlier sections had significantly longer mean nITIs compared to later sections. The interaction was also significant ($F(18, 288) = 3.91$, $p < 0.001$, $\eta_G^2 = 0.10$): the later sections had shorter nITIs for the none coupling condition only. For the SD nITI data, the main effect of Coupling was also significant ($F(3, 48) = 27.65$, $p < 0.001$, $\eta_G^2 = 0.22$) due to the significant reduction with increasing coupling. Similarly, the main effects of Section was also significant ($F(6, 96) = 15.68$, $p < 0.001$, $\eta_G^2 = 0.24$) as tap variability significantly decreased from earlier to later tap sections. Lastly, the interaction was significant ($F(18, 288) = 2.26$, $p = 0.0028$, $\eta_G^2 = 0.068$) indicated by coupling-related differences in tap variability changes over beat sections.

For Hybrid group, the main effect of Coupling was significant ($F(3, 69) = 165.26$, $p < 0.001$, $\eta_G^2 = 0.73$). The interaction between Coupling and Section was also significant ($F(18, 414) = 7.01$, $p < 0.001$, $\eta_G^2 = 0.11$). The post-hoc tests showed that the nITI significantly decreased over time for the none condition, while the opposite was true for the strong. For the nITI SDs, the main effect of Coupling was significant ($F(3, 69) = 18.86$, $p < 0.001$, $\eta_G^2 = 0.16$) as tap variability significantly decreased with increased coupling. The main effect of Section was also significant ($F(6, 138) = 10.25$, $p < 0.001$, $\eta_G^2 = 0.11$) as the tap variability significantly decreased over time. No interaction was found.

For Fast group, the main effect of Coupling was significant ($F(3, 24) = 126.16$, $p < 0.001$, $\eta_G^2 = 0.92$) as nITIs significantly decreased with weaker coupling. Also, the interaction between Coupling and Section was significant ($F(18, 144) = 2.80$, $p < 0.001$, $\eta_G^2 = 0.061$)

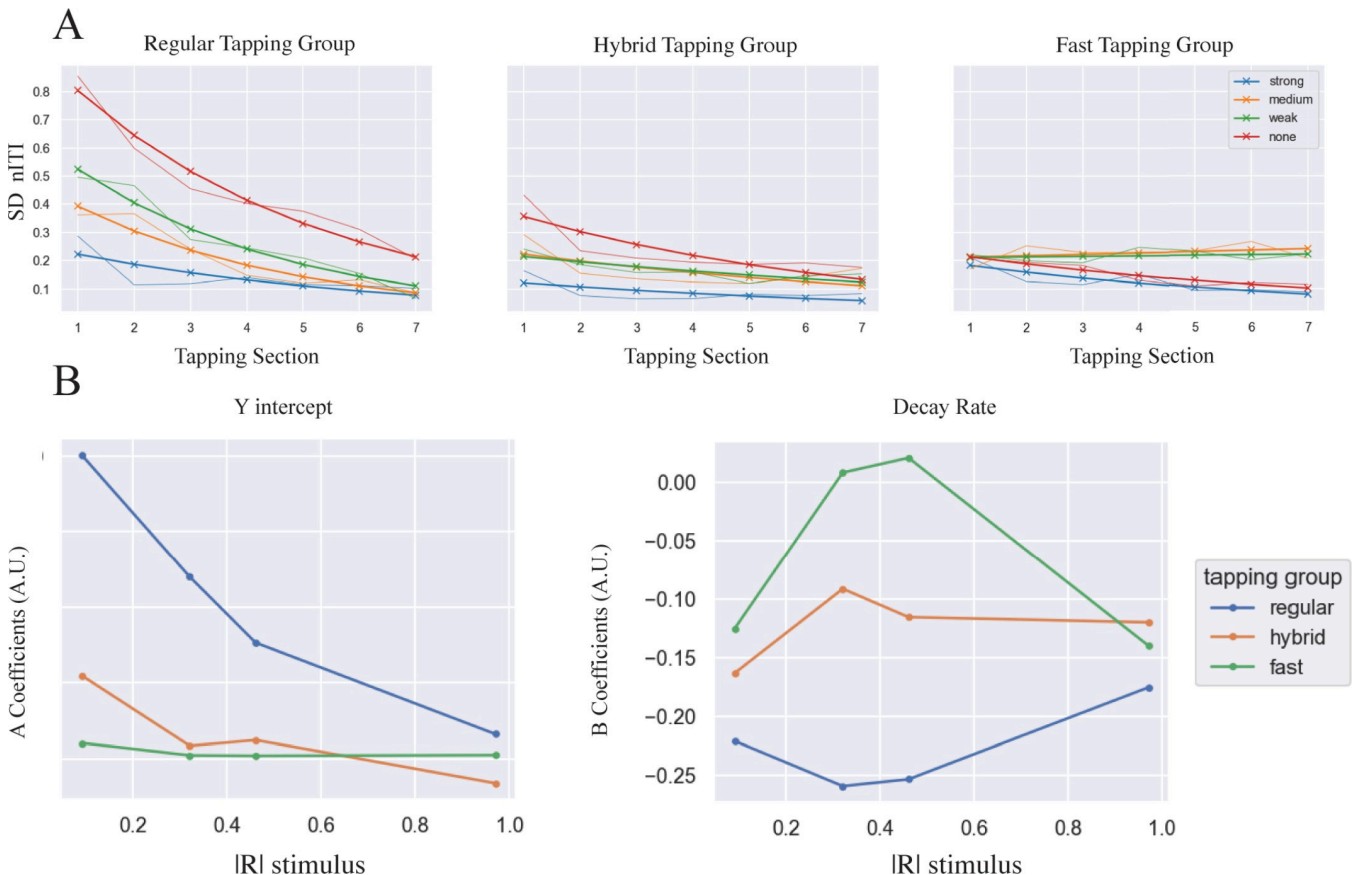

**Fig 5. Exponential curve fitting of standard deviation of nITI trajectory over tap sections.** A: The actual group mean data (thin) and the model using least squares polynomial fitting (bold), $y(t) = Ae^{B*t}$. B: Estimated A (intercept) and B (decay rate) coefficients over the phase coherence of the stimuli for each tapping group (details are in Table 2).

**Table 2. Results of exponential curve fitting.**

| Group | Coupling | A coeff. (p) | B coeff. (p) | $R^2$ |
|---|---|---|---|---|
| Regular | Strong | 0.27 (***) | -0.18 (0.08) | 0.53 |
| | Medium | 0.51 (***) | -0.25 (***) | 0.91 |
| | Weak | 0.68 (***) | -0.26 (***) | 0.94 |
| | None | 1.00 (***) | -0.22 (***) | 0.95 |
| Hybrid | Strong | 0.14 (***) | -0.13 (0.008) | 0.34 |
| | Medium | 0.25 (***) | -0.13 (0.02) | 0.38 |
| | Weak | 0.23 (***) | -0.09 (0.002) | 0.65 |
| | None | 0.42 (***) | -0.16 (***) | 0.67 |
| Fast | Strong | 0.19 (***) | -0.11 (0.005) | 0.58 |
| | Medium | 0.19 (***) | 0.04 (0.3) | 0.31 |
| | Weak | 0.21 (***) | 0.00 (0.5) | 0.00 |
| | None | 0.25 (***) | -0.13 (0.08) | 0.87 |

Curve fitting is performed to approximate the SD of nITI trajectories over tap sections, using $Ae^{B*t}$ where A coefficient is the intercept and B coefficient is the decay rate, indicated with p-value associated with the fit. The goodness of fit measure is the coefficient of determination $R^2$

('***', p < 0.0001).

reflecting that the later sections for the none coupling condition had significantly shorter nITIs. Tap variance remained small overall, but the main effect of Coupling was significant ($F$(3, 24) = 6.80, $p$ = 0.0018, $\eta_G^2$ = 0.26) due to the small variance for the none and strong coupling conditions. Lastly, the interaction was significant ($F$(18, 144) = 2.18, $p$ = 0.006, $\eta_G^2$ = 0.11) but the post-hoc test showed that the tap variability significantly decreased from the beginning to later sections only for the strong condition.

Fig 4C shows the phase coherence and tap-onset distribution plots. When examining the Rayleigh test of uniformity, with the exception of the none-coupled sounds, all of the coupled stimulus onsets had significant directionality ($ps < 0.001$). Similarly, all of the taps to the coupled (strong to weak) stimuli contained significant directionality ($ps < 0.001$). The Watson-Wheeler test revealed that the participants' tap and stimuli onset distributions per coupling condition were significantly different ($ps < 0.001$). Interestingly, the relationship between phase coherence magnitude across stimulus onsets and the one across participants' taps differed depending on the stimulus coupling. For the strong coupled stimuli, the stimuli phase coherence magnitude is larger than that of the taps regardless of tapping group. For Regular and Hybrid groups, the collective phase coherence magnitude was greater than that of the stimuli for the weak and medium coupling conditions and positioned at a slight lead. This means that the participants' taps as a whole were more aligned than the stimuli themselves.

## Characterization of frequent tapping: Delineating between dense and sparse tapping patterns

During the frequent tapping, some taps were made densely at a high frequency with low tap-to-tap interval variability, while others were made sparsely with more rhythmic variations. We suspected that these patterns are differently associated with the base tempo and the stimulus coupling as both affect how stimulus IOIs are distributed. If the dense pattern is used to express the fastest beats one can produce, then there would be no influence of the base tempo and no variability across subsequent tapping intervals. Conversely, people may use the sparse pattern to express the overall perceived beat pattern of stimulus. Therefore we used raw (un-normalized) ITI data from tapping data in cluster 3, and computed a measure of tap variability, the RMS dispersion and centroid. Then we split the data according to whether or not the dispersion is less than 0.1065: if so, the data were classified as 'dense', if not then they were classified as 'sparse'. For details on how these metrics are computed, please see the corresponding section in the S1 File. The mean raw ITI (*SD*) across all participants for the dense and sparse taps was 0.30 (*0.15*) and 0.37 (*0.15*) respectively, showing the clearly shorter interval for the dense pattern compared to the sparse ($p < 0.001$ according to a two-sample *t*-test). Furthermore, we found a group difference in the raw ITI mean value: the dense pattern raw ITI for the Regular, Hybrid, and Fast groups were 590, 480, and 211 ms respectively. These differences were significant (Regular vs. Hybrid (p = .011), Regular vs. Fast ($p < 0.001$), Hybrid vs. Fast ($p < 0.001$), by two-sample t-tests after Bonferroni correction). The sparse raw ITIs were 602, 453, and 305 ms for the respective groups (Regular vs. Hybrid (N.S.), Regular vs. Fast ($p < 0.001$), Hybrid vs. Fast ($p < 0.001$)). These results indicate that dense tapping is substantially more frequent than the sparse, and each group significantly differs in the raw ITI from other groups for both dense and sparse tapping except for the sparse tapping comparison between Regular and Hybrid groups.

We further assessed the proportion of the data for the dense and sparse tapping in each group for each of the four coupling conditions. Importantly, these proportion data per group are made for a descriptive purpose only, as the same data in the previous clustering procedure provided the criteria for classifying participants into groups. For the Regular group, the

original cluster 3 frequent tapping at 29% of the none coupled trials were separated into 15.0% of the dense and 14.4% of the sparse patterns. For the Hybrid group, the none coupled tap trials at 78% were separated into 37.5% of the dense and 40.8% of the sparse pattern. Lastly within the Fast group, for the medium and weak conditions, the dense patterns occurred comparably at 31.3% and 37.5%, respectively, but it jumped up to 70.0% at the none-coupled condition. The flip side of this is that they used the sparse tapping pattern for the medium and weak conditions for 52.5%, and 56.3%, but it dramatically reduced to 28.8% for the none condition (for more details, see the S2 Table).

We then applied multiple regression to test if phase coherence (R), stimulus tempo, and participant category predicted mean raw ITI within a trial, separately for dense and sparse tap trials. The model used is: mean raw ITI $\sim$ R*tempo*participant-category, assuming a maximal design with all fixed effect factorial interactions. The Regular tapping group was included in the base model as a reference for assessing the significance of group difference. For the dense pattern tapping, the overall regression was statistically significant ($R^2$ = 0.609, $F(11, 289)$ = 40.88, $p < 0.001$). The group assignment in Hybrid group ($B = 1.121$, $p = 0.010$) and Fast group ($B = 0.50$, $p = 0.007$) significantly predicted mean tap ITI for the dense trials differently from the base model with Regular group. However, there were no significant effects of coupling, tempo or interactions.

For the sparse pattern tapping, the overall regression was also statistically significant ($R^2$ = 0.559, $F(11, 195)$ = 22.43, $p < 0.001$). Here again, the group difference was significant for Hybrid group ($B = 0.650$, $p = 0.001$) and Fast group ($B = 0.641$, $p < 0.001$). Additionally, the interaction between stimulus tempo and the Hybrid group ($B = -0.006$, $p = 0.022$) and the stimulus tempo and the Fast group ($B = -0.007$, $p = 0.001$) was significant (for the complete regression results, please see S3 Table).

Thus, when people do arrive at dense-style tapping for dealing with weakly coupled stimuli, there is no effect of tempo or phase coherence– only the participant category had an effect. However, for the sparse tapping patterns, group categorization showed a significant interaction with the stimulus tempo, reflecting that these groups' tap responses were influenced by the underlying tempo. Notably, the difference between Regular and Hybrid groups for the sparse tapping was significant in the regression model, which was not the case by the simple t-test applied to the mean raw ITI.

## Discussion

This experiment was conducted with methodological improvements from Experiment 1 regarding sample size, stimulus sequence length, tempo variations, and the addition of stimuli without any explicit coupling. Having a larger cohort and longer sequences allowed us to evaluate the adaptive tapping behavior of participants better and more confidently. Extending the results in Experiment 1, we identified three participant groups, where each group had a substantial number of members who share common characteristics. This means none of the groups should be dismissed as outliers in the general population. The smallest Fast tapping group consists of 9 individuals out of 50, 18%. This number is similar when looking at fast tapping individuals in the Experiment 1 (2 out of 10). Importantly, all three groups shared both patterns of behavior, namely, regular beat tapping and highly frequent tapping. However, they differ not only in the level of reduced coupling at which they start to transition to the frequent tapping mode from regular beat tapping, but also the rate of tapping variability adaptation over time. Furthermore, the frequent tapping in the weaker coupling conditions is increasingly characterized with the dense tapping pattern where finger tapping is no longer influenced by

the coupling or the tempo, switched from the sparse pattern where groups showed distinct influences of the base tempo and its interaction with group membership.

This type of highly frequent tapping was also associated with less tap variability, consistent with the notion that tapping would become more regular upon increasing tempo and when approaching mechanical limits imposed by the human motor system [42]. The none-coupled stimuli included in this experiment provided an opportunity to see how participants may employ different tempo tracking strategies when confronted with sounds lacking tangible moments of concentrated sound density.

We will discuss more in detail how people switch strategies and what might determine individual differences in the General Discussion below.

## General discussion

Our tapping data revealed three main findings: (1) weaker coupling makes people switch to a frequent tapping mode and the criterion differs among individuals, largely separated into three groups (Regular, Hybrid, and Fast), (2) these groups also differed in terms of the tapping variability adaptation process as it occurred more slowly for the Regular group compared to the Hybrid and Fast groups even when everyone successfully performed regular beat tapping, and (3) group differences further explained tapping frequency variations, as the Fast groups exhibited predominantly dense tapping patterns which did not scale with tempo, while the Regular and Hybrid groups employed comparable proportion of dense and sparse tapping trials.

Our results confirm that strongly coupled stimuli clearly encourage regular beat extraction in all participants as we hypothesized. When the coupling becomes weaker, even though the majority of people still managed to extract regular beats, tap timing became more variable. This means that the spread in tapping could be viewed as a behavioral counterpart to the perceptual beat bin size which is gradually widening as the stimulus onsets were more widely spread, in line with the beat-bin model proposed by Danielsen [19, 20]. Most importantly, however, our novel findings also illustrate how the perceptual beat interval itself is processed differently across individuals. When stimulus coupling became extremely weak, people could no longer extract regular beats, and resorted to more frequent tapping. There was also a large inter-individual variability in terms of which level of coupling caused a departure from regular beat tapping. Our data further demonstrated that this transition is gradual for Regular and Hybrid groups: they had a sparse tapping pattern which scaled with the base tempo then increasingly switched to the dense pattern, but the dense pattern never became dominant. The transition of the Fast group, in constrast, appears almost immediate as they had predominantly showed the dense tapping pattern.

While the frequent tapping may be considered a simple breakdown or departure from the beat percept, we argue that the frequent tapping patterns still likely represents individuals' adaptive response to the stimuli's underlying onset structure rather than completely random or arbitrary individual idiosyncrasy. One strong indicator supporting the idea comes from the result where mean ITI in the frequent tapping scaled with tempo (i.e., the faster the tempo, the shorter the ITI became). This suggests that their taps still respond to the overall stimulus timing characteristics, while it is impossible to know whether people transition consciously. Furthermore, if participants completely lost the sense of beat, they could have stopped tapping altogether instead of performing the frequent tapping. One could suspect that participants may be resorting to preferred or spontaneous tapping rates when regular beats are not perceivable. However, this is unlikely because the mean ITI values in our data (for dense, 300 ms; sparse, 370 ms) are shorter than spontaneous rates documented in the literature, which typically cluster around 500–600 ms [3, 39]. Also, our data exhibited larger tap variabilities in

comparison to the expected stability associated with spontaneous rates. Based on these considerations, we propose that the genesis of the frequent tapping may reflect participants tracking the sequential surface level inter-onset intervals from oscillators. We refer this as a 'sequential' approach, in contrast with a more 'integrative' approach in which the collective onsets are used to derive the regular beat percept.

Such a 'shift' in the beat extraction mode might be related to how people integrate or segregate multiple auditory streams coming from rhythmic sources. Previous findings using temporal deviations are in line with this explanation and our data patterns. When temporal displacement around an isochronous beat (also known as "jitter") is sufficiently large, participants switched to tap at a faster rate, tracking the surface level rhythm formed from all streams rather than tracking each stream independently [43, 44], similar to our participants' frequent tapping. It is intriguing to see how the perceptual beat bin is first stretched for slightly weak coupling, followed by a subsequent reduction in the size of beat interval itself under even weaker coupling conditions. Even in our Regular group, their tap intervals in the none-coupling condition gradually became noticeably shorter over the course of the taps, potentially indicating the emergence of partial frequent tapping under the influence of time-sensitive adaptive mechanisms. Computational models of these systematic transitions involving both perceptual beat bin and beat interval size as well as their time course characteristics would be an interesting avenue for future research.

The tapping adaptation process differed between participant groups and was most noticeable when observing their tap-to-tap variability. All participant groups generally decreased this variability over time but the exponential curve fitting revealed the important difference between Regular and Hybrid tapping groups, especially with large differences between their intercept and decay rate parameters and how they interact with the stimulus coherence. The largest intercept and fastest decay rate in the Regular group indicated that their taps settled over a longer time period with more dynamic error correction. Also, the Regular group was more tolerant against the temporal spread of multiple onsets since they successfully integrated the stimuli compared to the other two groups, indicating that their perceptual beat bin may actually be wider. The Regular group members are more cautious and patient during their tap performance, making more moment-to-moment adjustments to the stimulus. The Hybrid tapping group also demonstrated a more integrative approach to the weakly coupled stimuli with a wider beat bin but had less tap variability overall across tap sections even when they managed to extract regular beats. In contrast, the Fast tapping group did not show much adaptation for their tap variability. This lack of adaptation might be explained by their frequent tap patterns, most of them as dense around the raw ITI of 250ms or less, the fastest among the three groups. It is known that tap consistency is proportionally related to tap rate, and that around the upper rate limits imposed by biomechanical constraints of SMS tasks, tap-to-tap inconsistencies are substantially reduced [45–47]. Thus, the low tap variability and absence of its adaptation change in the Fast group is likely the direct consequence of the very fast tapping. We speculate that these differential patterns in the adaptation process are also related to how their inner criterion to switch to frequent tapping operates. Given differences between perceived and measured synchrony [48], people may have different strategies when approaching uncertain and various timing intervals and their preferred way to explore the beat structure may also define how they adjust their tapping behavior over the course of the sequence.

These group differences in adaptation and frequent tapping prevalence may have resulted from the internal criteria and time constraints on how long they monitor the onset density (and/or interval variations) and how effectively they use those cues. Such criteria might be further informed by different prior experiences with or individual preferences for musical beats or metrical groupings [49–51]. These interact with stimulus characteristics such as tempo or

phase change, leading to critical aspects of music performance and predictive behaviors. However, it is important to note that our group categorization was not related to musical training or other demographic characteristics.

What underlies the sparse tapping behavior? One idea might be that people try to mimic, track, and/or emulate the sound onsets that are distributed rather irregularly. The proportional relationship between tap frequency and tempo in the sparse tapping observed in our results supports this idea. Previous SMS studies have also shown how participants tend to echo a rhythmic sequence if the sequence is too unpredictable, exhibiting a large positive lag-1 cross correlation [52–54]. Similar to techniques approximating the time-averaged statistics for faithfully recreating the sound texture of water or insects swarming [55], time estimation central tendency occurs in perception by deriving a Bayesian prior from statistical estimate from an IOI distribution [56], consistently shown in auditory and visual modalities [57, 58]. Also, such tracking behavior in embodied musical cognition could entail imitative gestures, movements, and behaviors that shape temporal expectancy [59, 60]. Indeed, movements during silence facilitated timekeeping easily in all the conditions with some degree of coupling [61, 62]. Thus, the sparse tapping might be part of a motor process to express perceived present and seek matching by active participation.

Our task and parametric approaches revealed nuanced and systematic differences in the spectrum of tapping responses, where all individuals showed all of the tapping categories but with different proportions. Our Fast group, with the smallest number of people, switch to frequent tapping at the medium coupling, whereas the rest of the participants who were part of Regular or Hybrid groups (above 80%, 8 out of 10 in Exp1, 41 out of 50 in Exp2) still found regular beats easily. This might be related to the documented 'beat deaf' individuals in the literature. Such individuals do not generally have difficulty synchronizing their taps to isochronous single pacing signals with a metronome but they often cannot detect beats in more complex, musically-oriented or metrically-structured stimuli, and show difficulties in responding to perturbations in otherwise isochronous patterns [63–65]. Noteworthy is the consistency of the proportion (15%-17%) of such poor synchronizers in the general population of young adults [31, 32] with our Fast tapping group of 18%.

Research into spontaneous synchronization in speech has also shown two groups characterized as exhibiting 'high' or 'low' synchronization in their ability to align syllables with rhythmic sequences [66]. Since we did not use music, speech, or metronome with perturbations, future research should further investigate the potential link between our data and these findings. Spontaneous rates may also account for some of the individual differences we observed. Spontaneous tapping rates vary across individual [67], and non-musicians tend to exhibit less flexibility in synchronizing to a range of tempi and rhythmic patterns, biasing their tap response to their intrinsic frequency [3, 68]. Particularly relevant to our observation is the finding that the difference between two people's spontaneous rates influences how stably they can perform the joint tapping [69, 70], which also relates to the coupled oscillator models [22–24, 71].

These individual differences may arise from different functionality in the brain structures responsible for time perception and action integration, such as cerebellum, striatum, supplementary motor area and prefrontal cortex [72]. While striato-thalamo-cortical circuits are considered to contribute to beat-based timing and maintenance, the cerebellar system mainly supports timing adaptation processes. Patients with cerebellar lesions show deficits in temporal coordination in motor (and non-motor) behavior with wider distributions of spontaneous frequency, increased negative mean asynchrony when tapping to isochronous stimuli, as well as a compromised ability to detect and respond to stimuli perturbations [73]. Thus, it is possible that many individual differences in tap strategy are caused by differences in processing temporal, event-based sensory input and how error correction and adaptation operate. Indeed,

individuals have different capacities for adaptability when responding to auditory environments in the absence of explicit sensory feedback, which may be a qualitative measure of health [74]. Interestingly, a recent study reports genetic variation that can explain beat perception and musicianship [75] but the findings in this area of research have yet to converge [76]. The novel timing process modes within individuals observed here, to our knowledge, has not been examined in the literature and presents a number of possible avenues for uncovering its neuro-anatomical underpinnings and related sensorimotor/cognitive functions.

Online testing platforms such as one used here are disadvantageous for psychophysical adaptive procedures [77]. However, it is important to emphasize that our primary measures such as frequency of the taps and phase distributions are not influenced by these shortcomings in the experimental apparatus. Since we did not collect any data on the participants' preferred or spontaneous tapping tempo, we have no way to explore whether or not people used a certain tapping rate or preferred tapping pattern when they fail to hear an underlying beat in the stimulus.

Despite the technical limitations, our analysis was robust by the predominance of our ITI based analysis methods. Our experimental prompt was left intentionally vague and open-ended so as to not encourage a priori assumptions about how they should tap to the stimuli. A more rigorous analysis of tapping pattern beyond the phase portrait might allow us to gain further insight into the participants' level of entrainment. For example, recent research has applied phase-space recurrence plots, normalized pairwise variability index (nPVIs), correlation methods, and pulse generate-and-test (GAT) analysis in order to look at temporal and rhythmic structure in drumming performance, human and animal vocalizations [78, 79]. The low temporal predictability found in many of the weaker-coupled stimuli, may have encouraged participants to adopt more complex approaches to tapping.

## Conclusion

The present study examined how the perception of beat arises in an auditory scene comprised of phase-coupled oscillators which served as a proxy for large-scale, interactive systems found in music, ambient environments, and other natural phenomena more broadly. Our data confirmed our hypothesis regarding inter-individual variability in beat extraction approaches; we uncovered the novel phenomena of frequent tapping modes people use when density cues are weak, and how people vary at which degree of coupling they lose the perceptual density cues to entrain and connect to the collective environment. Our results also confirmed our first hypothesis in that when the coupling is stronger, people show integrative processing of multi-onset information to achieve regular tapping. As long as the participants can maintain the regular beat extraction mode, the variance of inter-tap intervals increases as the coupling becomes weaker. In these weakly coupled stimuli, sound onsets are more widely distributed across beat centers and therefore we would expect a consequent widening of the perceptual-beat bin. The adaptive behavior was also captured in our data, confirming our second hypothesis. Importantly, the adaptive trajectory varied across the identified groups differing the transition point to the frequent tapping. The frequent tapping likely reflects both the sequential processing mode which may be dictated by the available interval size for perceived beat, and the active participation/imitation approach for searching adjustments. Thus, individual transition point between integrative and sequential processing appears linked directly and systematically to inner criteria about timing processing windows and adaptation. The coupling parameters of the generative model provides a useful systems approach to further investigate how people differ in their ability to consciously or unconsciously coordinate with each other and the natural environment.

## Supporting information

**S1 Fig. Experimental flow chart.** Left top panel ('STIMULUS GENERATION'): Kuramoto model was used to generate audio stimuli with 40 coupled oscillators with different coupling strength conditions (strong, medium, weak for Exp.1 and strong, medium, weak, none for Exp. 2) and various base tempo (90–110 BPM for Exp.1 and 72–119 BPM for Exp.2). Left bottom panel ('DATA COLLECTION'): The generated audio stimuli were used for tapping data collection. Right panel ('DATA ANALYSIS'): Kuramoto model behaviors were used to determine beat windows which were compared against tap timing, resulting in the phase coherence regarding tapping onsets, stimulus onsets, and tap and stimuli relationships. Inter-tap intervals were also analyzed to see its frequency compared to the beat bin size, and adaptive patterns over stimulus sequence duration in a given trial. For more information see S1 File.
(TIF)

**S2 Fig.** (Left) Histogram of tap dispersion. The red vertical line indicates the criterion to separate 'dense' and 'sparse' tap responses. (Right, top row) Stimulus wave form (blue) with beat windows (orange), beat centers (green), and a sample individual tap responses (red). (Right bottom row). Examples of "phase portraits" for the intervals of each subject's tap response (red) and stimulus beat centers (blue) for a tap response part of the cluster 1 as regular beat tapping (A.) and two frequent tap responses part of the cluster 3 that was classified as 'dense' (B.) and 'sparse' (C.) respectively.
(PDF)

**S1 Table. Results of the two-way ANOVAs and post-hoc tests for the mean and SD of the nITI for the regular, hybrid, and fast tapping groups in Experiment 2.** Note. Significance levels are indicated as *** $p < .001$, ** $p < .01$, * $p < .05$, ns = not significant. For post-hoc results: S = strong, M = medium, W = weak, N = none, T1-7 = tapping section 1–7, >>> = significantly greater ($p < .001$), >> = significantly greater ($p < .01$), > = significantly greater ($p < .05$), $\sim$ = not significantly different.
(DOCX)

**S2 Table. Percentages of trials and (SD) per subject group for 'dense' and 'sparse' tap responses within the frequent tapping in the cluster 3.**
(DOCX)

**S3 Table. Results of regression analyses as applied to mean ITI and dispersion for dense and sparse tap responses in the cluster 3.** Note. Significance levels (S) are indicated as *** $p < .001$, ** $p < .01$, * $p < .05$.
(DOCX)

**S1 File. Experimental flow chart, beat window segmentation of the auditory stimuli, characterization of frequent tapping: Delineating between dense and sparse tapping patterns, effect of tempo and group on the raw ITI and dispersion: Multiple regression.**
(DOCX)

## Acknowledgments

The authors sincerely thank Noah Fram and Aditya Chander for their helpful suggestions to the earlier version of the manuscript.

## Author Contributions

**Conceptualization:** Nolan Lem, Takako Fujioka.

**Data curation:** Nolan Lem, Takako Fujioka.

**Formal analysis:** Nolan Lem, Takako Fujioka.

**Investigation:** Nolan Lem, Takako Fujioka.

**Methodology:** Nolan Lem, Takako Fujioka.

**Resources:** Nolan Lem.

**Software:** Nolan Lem.

**Visualization:** Nolan Lem.

**Writing – original draft:** Nolan Lem.

**Writing – review & editing:** Nolan Lem, Takako Fujioka.

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
