## [Decision Letter · Decision Letter 0]

24 Nov 2022

PONE-D-22-25174

Individual differences of limitation to extract beat from Kuramoto coupled oscillators: Weak coupling induces emergence of frequent tapping

PLOS ONE

Dear Dr. Lem,

Thank you for submitting your manuscript to PLOS ONE. After careful consideration, we feel that it has merit but does not fully meet PLOS ONE’s publication criteria as it currently stands. Therefore, we invite you to submit a revised version of the manuscript that addresses the points raised during the review process.

We look forward to receiving your revised manuscript.

Kind regards,

Alice Coles-Aldridge

Editorial Office

PLOS ONE

Journal Requirements:

3. Please change "female” or "male" to "woman” or "man" as appropriate, when used as a noun (see for instance https://apastyle.apa.org/style-grammar-guidelines/bias-free-language/gender).

5. Please ensure that you include a title page within your main document. You should list all authors and all affiliations as per our author instructions and clearly indicate the corresponding author.

Additional Editor Comments:

The manuscript has been evaluated by two reviewers, and their comments are available below.

The reviewers have raised a number of major concerns. They feel some aspects of the subtitles, abstract, discussion and conclusion could better reflect the main findings, and that the presentation and discussion of the theoretical framework provided by the beat bin model could be more in-depth. 

Could you please carefully revise the manuscript to address all comments raised?

Reviewers' comments:

Reviewer's Responses to Questions

**Comments to the Author**

1. Is the manuscript technically sound, and do the data support the conclusions?

Reviewer #1: Yes

Reviewer #2: Yes

2. Has the statistical analysis been performed appropriately and rigorously? 

Reviewer #1: Yes

Reviewer #2: Yes

3. Have the authors made all data underlying the findings in their manuscript fully available?

Reviewer #1: Yes

Reviewer #2: Yes

4. Is the manuscript presented in an intelligible fashion and written in standard English?

Reviewer #1: No

Reviewer #2: No

5. Review Comments to the Author

Reviewer #1: This is a very interesting study that presents original research regarding how different groups of listeners extract regular beats from multiple streams of sound onsets. Using a dynamical system of Kuramoto oscillators the authors simulate different degrees of loose timing, from tightly coupled streams affording almost metronomic precision to weakly coupled streams affording no clear beat, and have people tapping to the different conditions. The results show that the tapping variability, presumably reflecting the perceptual beat bins in the tappers, increases when the coupling between the oscillators weakens, to the point where it breaks down altogether. The participants cluster in three groups with varying beat extraction abilities.

Experiments, statistics and other analysis are sound. The findings are supported by the data. However, the subtitle (Weak coupling induces emergence of frequent tapping) and some aspects of the abstract, discussion and conclusion are slightly misleading and should be rephrased to better reflect the main findings. Moreover, some potentially interesting aspects of the study are underilluminated in the current version of the manuscript.

The claims are placed in context of the previous literature, but the presentation and discussion of the theoretical framework provided by the beat bin model, which is key to the study, is somewhat superficial.

The structure of the manuscript is clear, and the text is for the most part intelligible, but some parts of the manuscript need to be improved as to both clarity and linguistic quality.

For details and suggestions, see list of major and minor points below.

Major comments:

1. The authors point to the beat bin model as highly relevant to the authors' main question regarding how different groups of listeners extract regular beats from multiple streams of sound onsets. The presentation of the beat bin model is, however, not very thorough and slightly imprecise. On p. 4, for example, the beat bin model is presented as follows: “The beat bin model was originally conceived to explain how groove-based musical genres, where music is typically following a steady tempo constraint, might still allow listeners to feel the stretch of beats.” This is kind of true, but also a bit confusing since the argument rather goes in the opposite direction, that is: Also in music with stretched (extended) beats, we can feel a steady tempo. In the abstract to Danielsen 2018 (see full reference below), for example, the rationale behind the beat bin model is presented as follows: “multiple pulse locations change the shape of the beats [that is, the beat bins in the signal] and also affect the internal beat [the perceptual beat bin] that the listener uses to make sense of the rhythmic events. One way to produce extended beats is to introduce multiple pulse locations at the micro level of a groove.” Applied to the current study, this means that the beat bins in the signal (formed by the density of onset peaks) is hypothesized to influence the perceptual beat bins (measured as tapping variability). Generally, the more conceptual and theoretical aspects of the beat bin model could have been more elaborated and also more actively used in the discussion and conclusion. This would have improved the coherence of the manuscript. As it reads now, the beat bin model is introduced early on as an important concept, but then mostly used as a label for the beat width computed by the beat segmentation algorithm.

2. One finding should be given more consideration, namely the increase in the spread in tapping (higher variability) when the coupling gets weaker and the beat bin in the signal increases. This result is obvious from the data and entirely in line with the beat bin model, but not highlighted in the manuscript. It is interesting because tapping, as pointed out above, is often considered an explication of the subjective beat or inner pulse that a listener extract from the incoming stimuli. Given this, the spread in tapping could be considered a behavioural counterpart to the perceptual beat bin (this is how beat bin is operationalised in the study by Danielsen et al., 2019).

3. On that note, it might be useful to distinguish clearly between the beat bins that beat segmentation algorithm extracts from the stimuli and the (perceptual) beat bin that the tapping behaviour is thought to reflect. One could do this by naming the computational beat bin beatbincomp and the perceptual beat bin, as reflected in the tapping variability, beatbinpercept or simply beatbintap, for example. Generally, it should be clear whether one refers to the oscillator model/signal or to perception/tapping when beat bins, spread, variability etc are being discussed.

4. In the general discussion the authors sum up one of the main findings as differences between groups regarding when they switch to what they call “the frequent tapping strategy”. (This is the finding alluded to in the subtitle). Labeling this a strategy seems, however, slightly misleading. Rather, the frequent tapping strategy seems to be what participants do when losing the feeling of a regular beat, that is, when they are no longer able to extract a beat from the stimuli. I thus suggest finding a different name for this, such as, for example, “irregular tapping.” In the abstract, you suggest that this because of different perceptual timescales in the participants. In my view, this is not very likely, and the fast tapping should be rather discussed as a break-down in beat-extraction ability. It also comes forward as weird to foreground faster tapping variability adaptation for the Hybrid and Fast groups compared to the Regular as a main finding given that the former groups probably adapt to their own pace and not to the periodicity of the stimuli. The statement regarding how “the Regular group employed comparably low proportion of dense and sparse tapping trials” testifies to this, as this means that this group in fact tapped in accordance with the periodicity of the stimuli. The focus of this first paragraph of the general discussion thus strikes me as a sidetrack, and I suggest rewriting it with the main hypothesis and the theoretical framework in mind.

From the second paragraph and onwards the general discussion is much more to the point and here it also becomes clear that the frequent tapping “strategy” in fact is another name for losing the ability to extract a beat from the stimulus, as is clear from the following: “when stimulus coupling was weaker, people resorted to more frequent tapping, revealing individual differences at which level of coupling to do so. This departure from regular beat tapping indicates that people could no longer rely on the dynamics of the sound onset density change in order to extract beat.» The identification of different breakdown levels for beat extraction for the three groups is a major finding. However, by presenting fast tapping as a strategy rather than inadequate tapping, the authors are obscuring this interesting point. In general, too much attention is given to frequent tapping as such. I would say that the only interesting thing here is at what coupling level the shift happens for the different groups.

5. The structure of the conclusion should be improved. Please revisit your main hypothesis and sub-hypothesis and address the relevant results point by point.

Minor points:

p. 4 ‘Sound onset density’ is introduced before the method is presented and it is not immediately clear what this means at this point in the manuscript. Please explain.

p. 4: “In that sense, each individual may serve another agent with a coupling factor, contributing to the whole system.” Unclear, please explain.

p. 5: “This predicts that stronger coupling conditions would result in more consistent tapping intervals and phase relationship to other participants compared to weaker coupling conditions.» Here one might get the impression that several participants are tested simultaneously.

p. 9: “This resulted in a total of 30 audio stimulus files.” I was a bit confused here. Isn’t the correct number 60: 3 coupling strengths x 5 tempi x 4 versions = 60?

p. 9-10: Using the space bar on a laptop is not the best solution for obtaining the millisecond precision needed for research into microtiming. I appreciate that the authors discuss potential limitations related to latency issues and provide the technical specifications of their equipment towards the end of the section “Test Procedure”. However, this part of the manuscript is somewhat difficult to follow and would benefit from more clarity.

p. 10, several unclear sentences:

“The generative coupled oscillator model produced stimuli using different coupling strengths. to distribute sound onsets different periodic moments in time.” This sentence is not clear and feels incomplete. Please rewrite.

“Briefly, the generated stimuli’s average angle, � at each time step was used to determine beat bins using the beat segmentation algorithm extracted the referent beat centers.” Incomplete sentence. Missing comma after step?

“...circularly mapped onto the beat bins in order to perform phase coherence analysis (per beat region)…” This is an example of beat bin referring to the signal. This should be made clear.

«For ITI analysis, we are concerned the amount of time (period)…» Unclear. Please fill in missing word(s).

p. 16: What kind of musical training? Formal or practice or both?

p. 27: “…significantly larger for the strong coupling condition compared to the weaker coupling conditions…” Larger or perhaps ‘longer’?

p. 31: In the first sentence of the second paragraph of the general discussion (p. 31), the authors state: “Our results confirm that strongly coupled stimuli clearly encourages regular beat extraction in all participants, supporting the idea of ‘beat bin’ (Danielsen, 2010; Danielsen et al., 2019) as we hypothesized.» As this reads now, it is a bit unclear whether “as we hypothesized” refers back to the first or second part of the sentence. It should also be noted here that the beat bin model predicts that narrow bins in the signal, or strongly coupled stimuli, will encourage regular beat extraction in the sense of both stable period synchronization and tight phase synchronization. Wider bins in the signal will produce wider perceptual bins, that is, looser phase synchronization and more tapping variability. As mentioned above, the latter finding is important and should be listed among the main findings.

p. 32: “The largest intercept and fastest decay rate in the Regular group indicating their taps were settled over a longer time period with more dynamic error correction.” Correct the syntax. It should also be noted here that this could be because of a general inclination towards more narrow beat bins in this group.

“…if people have different strategies in approaching towards the uncertain and various timing intervals…” When approaching or In their approach to.

Regarding the possibility of different strategies, it might be that the wider perceptual beat bins of the hybrid group facilitate stable periodicity because more taps will be experienced as within the beat bin when the bins are wide. See Matthews et al. 2022 for an interesting article on the relationship between perceived and measured synchrony.

p. 33: “Altogether, our paradigm provides a new way to assess individual tapping strategies and adds to the current knowledge of beat induction processes that are informed by top-down predisposition/preferences and bottom-up stimulus-driven processes.» How? Please unpack this claim.

In the discussion, it is sometimes unclear whether you refer to other studies or your own work. Be sure to make this clear by adding “previous research has shown… “ etc.

References:

Danielsen, A. (2018). Pulse as dynamic attending: Analysing beat bin metre in neo soul grooves. In The Routledge Companion to Popular Music Analysis (pp. 179-189). Routledge.

Matthews, T. E., Witek, M. A., Thibodeau, J. L., Vuust, P., & Penhune, V. B. (2022). Perceived Motor Synchrony With the Beat is More Strongly Related to Groove Than Measured Synchrony. Music Perception: An Interdisciplinary Journal, 39(5), 423-442.

Reviewer #2: I read this article with much interest Interindividual differences in tapping to auditory stimuli with various timing properties. The main finding that participants use different strategies to coordinate their movement with more or less salient beats is very interesting and contributes to our understanding of humans’ rhythm processing skills.

The writing of the paper should be improved. There are frequent grammatical mistakes (e.g., “Our results confirm that strongly coupled stimuli clearly encourages”, p.31). Numerous examples of unclear sentences can be found throughout the manuscript. I strongly encourage the authors to have the article proofread by several people. Also, the length of the paragraphs (almost 1.5 pages for some of them in the discussion section) is too long, making it particularly difficult to follow. For example, page 34 should be divided into three paragraphs, focusing on beat deafness, spontaneous production rates, and neuronal bases, respectively.

p5, §2 “to other participants” (towards the bottom of the page): is that correct? This suggests that this a question address by a cross subject design

p6, §1 “might employ different behavioral patterns, related to their internal tendencies

that interact with external sounds of different coupling strength” (towards the top the page): it is not clear what behavioral patterns and internal tendencies mean here, and it does not seem that they are explicitly explained earlier in the introduction.

There are some points that are unclear to me in the hypotheses formulated. It should be specified that coupling, here, refers to the coupling between the different oscillators generating the sounds. It is particularly important because it is not easy to understand that the stimuli consisted in clustered sounds with a “beat” at the peak level metronome sequences (i.e., single sounds) at a quasi-periodic frequency. This should be clarified in the introduction. It would help to include a schematic of the stimuli in the strong, medium, weak, and none phase coherence conditions:

Strong: I I I I I I

Weak: I I I I I I I I

It becomes clearer towards the results and discussion sections stimuli are actually clustered sounds. Individual dots representing the sound of each oscillator could be added above the bars.

I would appreciate if the authors could provide example stimuli in the next submission.

Experiment 1

I am not convinced that presenting this first experiment is necessary. It is not common to present pilot experiments, and the added value here seems very small. The article is dense, with very complex analysis. I think it would be much easier to read, and the main results will be conveyed more efficiently if this experiment is not presented. The authors can mention in the introduction that a short-scale, in-person pilot study using a smaller range of stimuli tempi and duration was conducted prior to the experiment to show that their design capture people’s naturalistic tapping behavior.

Experiment 2

Why did the authors use the nITI instead of the mean asynchrony, or the vector angle? I understand after reading the discussion that it is because the relative timing is not reliable with the experimental setting. This should be explained earlier in the article. Note that this increases significantly the complexity of the analyses.

There are some repetitions across the methods and the results section, especially for the clustering analyses.

Please report effect sizes for statistical analyses when relevant.

The results second paragraph of page 24 (beginning with “we examined”) could be summarized in a table to reduce the text length

Page 27: the wording of the presentation of the ANOVA is awkward (The main effect (…) was significant because the post-hocs show)

Also, why was the Group not included as an independent variable in the ANOVA, instead of running the same anova on the 3 different groups? I expect an overall difference between the 3 groups by eyeballing figure 4A

The order of the figures is confusing. For example, figure 3C is presented after figure 4 in the text.

Page 28, paragraph 3 (beginning with “Interestingly”). What analyses this interpretation correspond to? Are they supported by statistical analyses?

In general, the authors could be more cautious with interpretation of results not supported by inferential statistics (see also p25, last paragraph), or at least explain better why these interpretations are performed. Again, there are a lot of results and analyses that are very difficult to grasp for the reader. Focusing on the main findings would definitely help.

Discussion

“Frequent tapping strategy” does not seem like an appropriate way to say that people tap faster than stimulus tactus.

Regarding spontaneous production rates. A recent paper showed the influence of people’s spontaneous rates of production in the context of non-linear dynamic modeling (Bégel, Demos, Wang, & Palmer, 2022).

“tracking the collective rhythm rather than tracking separate streams” (p31, first paragraph). The authors should clarify what they mean by collective rhythm.

Page 33, middle paragraph. I don’t understand the argumentation. It seems to me that the paragraph covers different points, and it is not clear how they are related to the study, especially because these points are not presented in the introduction. I also miss the general idea here. What does “short stimulus onset intervals” mean? I understand that it’s the interval between each oscillator, but this is very confusing.

Minor comments

When referring to humans, “Female” should be employed as an adjective. Although the formulation used by the authors (5 females) is frequent in scientific articles, my opinion is that another noun should be used instead (women), or female should be employed as an adjective, as it can be perceived as offensive (the definition of Wikipedia clearly states “now possibly offensive in certain contexts”). I am not a native English speaker, but it seems to me that using the singular form of female in the sentence would turn it into an adjective implicitly related to the word “volunteers” and therefore be grammatically correct: “10 volunteers from the Stanford University community took part in this study (5 female, M age = 31 years, SD = 5.6 years, all right handed).”

It also seems to me that contracted forms (“It’s”, p8 §1) should not be used

6. PLOS authors have the option to publish the peer review history of their article (what does this mean?). If published, this will include your full peer review and any attached files.

Reviewer #1: **Yes: **Anne Danielsen

Reviewer #2: **Yes: **Valentin Bégel

---

## [Author Response · Author response to Decision Letter 0]

29 May 2023

Dear reviewers,

We appreciate your detailed and insightful comments. We think that our manuscript has improved by taking into account your criticisms and suggestions. Also, as we communicate to the editor, alongside the revision process, we organized the raw data for anyone interested for downloading. Alongside, we updated part of figures and statistics because we eliminated two samples previously duplicated due to a technical error we discovered (Exp2, Hybrid group). This update did not change any of the result patterns or interpretation - group designation, model fitting, or which test was significant in each of the statistical analyses. Below we respond to each of your comments and note where and how those responses are handled in the revised manuscript. Revisions to the manuscript and our responses to comments below are in blue font. 

Reviewer #1: This is a very interesting study that presents original research regarding how different groups of listeners extract regular beats from multiple streams of sound onsets. Using a dynamical system of Kuramoto oscillators the authors simulate different degrees of loose timing, from tightly coupled streams affording almost metronomic precision to weakly coupled streams affording no clear beat, and have people tapping to the different conditions. The results show that the tapping variability, presumably reflecting the perceptual beat bins in the tappers, increases when the coupling between the oscillators weakens, to the point where it breaks down altogether. The participants cluster in three groups with varying beat extraction abilities.

Experiments, statistics and other analysis are sound. The findings are supported by the data. However, the subtitle (Weak coupling induces emergence of frequent tapping) and some aspects of the abstract, discussion and conclusion are slightly misleading and should be rephrased to better reflect the main findings. Moreover, some potentially interesting aspects of the study are underilluminated in the current version of the manuscript.

The claims are placed in context of the previous literature, but the presentation and discussion of the theoretical framework provided by the beat bin model, which is key to the study, is somewhat superficial.

The structure of the manuscript is clear, and the text is for the most part intelligible, but some parts of the manuscript need to be improved as to both clarity and linguistic quality.

Thank you for the overview of the evaluation and positive comments on the originality and the significance of our study. As described below, we carefully revised our manuscript to clarify the issues raised. With respect to the subtitle, we still think that the transition from beat-based tapping to frequent tapping with weaker coupling is part of the major findings here. As such, we revised our subtitle as ‘transition from beat-based tapping to frequent tapping with weaker coupling.’ Relatedly, throughout the revised manuscript, we made further discussion on how to interpret our data not only in the light of the perceptual beat-bin size but also in the light of the relationship between the stimulus and tapping interval sizes. 

1. The authors point to the beat bin model as highly relevant to the authors' main question regarding how different groups of listeners extract regular beats from multiple streams of sound onsets. The presentation of the beat bin model is, however, not very thorough and slightly imprecise. On p. 4, for example, the beat bin model is presented as follows: “The beat bin model was originally conceived to explain how groove-based musical genres, where music is typically following a steady tempo constraint, might still allow listeners to feel the stretch of beats.” This is kind of true, but also a bit confusing since the argument rather goes in the opposite direction, that is: Also in music with stretched (extended) beats, we can feel a steady tempo. In the abstract to Danielsen 2018 (see full reference below), for example, the rationale behind the beat bin model is presented as follows: “multiple pulse locations change the shape of the beats [that is, the beat bins in the signal] and also affect the internal beat [the perceptual beat bin] that the listener uses to make sense of the rhythmic events. One way to produce extended beats is to introduce multiple pulse locations at the micro level of a groove.” Applied to the current study, this means that the beat bins in the signal (formed by the density of onset peaks) is hypothesized to influence the perceptual beat bins (measured as tapping variability). Generally, the more conceptual and theoretical aspects of the beat bin model could have been more elaborated and also more actively used in the discussion and conclusion. This would have improved the coherence of the manuscript. As it reads now, the beat bin model is introduce¬d early on as an important concept, but then mostly used as a label for the beat width computed by the beat segmentation algorithm.

Thank you for the suggestions on this critical point on the beat bin model. 

In this revision, we carefully revised the relevant parts about the Danielsen’s model. In the introduction, we explained the model more rigorously to avoid misrepresentation, so that the readers could understand how tap variability relate to the P-center and the spread of the beat. 

In the discussion, we pointed out explicitly that this model is in line with our data as long as participants were able to integrate various intervals and extract regular beat, characterized by the increased tap variability (thus stretching the perceptual beat-bin) with weaker stimulus coupling. However, when the coupling becomes further weak, participants no longer integrate the multiple intervals to extract regular beat and resort to ‘frequent tapping’ where their tapping intervals gradually shrink from ‘sparse’ to ‘dense’ styles. We thus propose that the observed transitions and group differences are associated with the perceptual beat ‘interval’ size, once people switch from an integrative approach to the sequential approach to process short intervals. We hope our added explanation in the Introduction and Discussion make clear our interpretation of the data and the relationship to the beat-bin model. 

2. One finding should be given more consideration, namely the increase in the spread in tapping (higher variability) when the coupling gets weaker and the beat bin in the signal increases. This result is obvious from the data and entirely in line with the beat bin model, but not highlighted in the manuscript. It is interesting because tapping, as pointed out above, is often considered an explication of the subjective beat or inner pulse that a listener extract from the incoming stimuli. Given this, the spread in tapping could be considered a behavioural counterpart to the perceptual beat bin (this is how beat bin is operationalised in the study by Danielsen et al., 2019).

Thank you for this helpful comment. As in our response to the previous comment, we agree that the perceptual beat bin is effectively widened upon weaker coupling, indicated by the spread of onsets made tap variability increase in our data. However, the observed frequent tapping pattern for the even further weaker coupling conditions indicates the diminished beat-to-beat interval size, rather than the infinitely elongated beat-bin size. These points are now articulated in the revised discussion (p. 27-28).

3. On that note, it might be useful to distinguish clearly between the beat bins that beat segmentation algorithm extracts from the stimuli and the (perceptual) beat bin that the tapping behaviour is thought to reflect. One could do this by naming the computational beat bin beatbincomp and the perceptual beat bin, as reflected in the tapping variability, beatbinpercept or simply beatbintap, for example. Generally, it should be clear whether one refers to the oscillator model/signal or to perception/tapping when beat bins, spread, variability etc are being discussed.

Thank you for pointing this out. We have replaced the terms to avoid the confusion. Specifically we now call the perceptual beat bin as ‘beat bin’ and call beat segmentation algorithm as ‘beat window segmentation algorithm’ that determines ‘Beat window’ throughout the manuscript.

4. In the general discussion the authors sum up one of the main findings as differences between groups regarding when they switch to what they call “the frequent tapping strategy”. (This is the finding alluded to in the subtitle). Labeling this a strategy seems, however, slightly misleading. Rather, the frequent tapping strategy seems to be what participants do when losing the feeling of a regular beat, that is, when they are no longer able to extract a beat from the stimuli. I thus suggest finding a different name for this, such as, for example, “irregular tapping.” In the abstract, you suggest that this because of different perceptual timescales in the participants. In my view, this is not very likely, and the fast tapping should be rather discussed as a break-down in beat-extraction ability. It also comes forward as weird to foreground faster tapping variability adaptation for the Hybrid and Fast groups compared to the Regular as a main finding given that the former groups probably adapt to their own pace and not to the periodicity of the stimuli. The statement regarding how “the Regular group employed comparably low proportion of dense and sparse tapping trials” testifies to this, as this means that this group in fact tapped in accordance with the periodicity of the stimuli. The focus of this first paragraph of the general discussion thus strikes me as a sidetrack, and I suggest rewriting it with the main hypothesis and the theoretical framework in mind.

This is a very insightful comment that helps improving our paper.

First of all, we think that the observed fast tapping pattern is still a very important part of our findings, because three groups showed the sparse and dense patterns when the regular beat extraction is increasingly difficult. This indicates something more systematic in its underpinning mechanism rather than simply ‘irregular.’ 

Nevertheless, we agree with the reviewer that whether one could call this as a strategy is unclear, because we do not know whether participants made a conscious decision. 

In fact, it is not possible for us to determine what participants are thinking besides their tapping behavior – they may have simply felt a faster beat, completely lost the sense of beat, entrained to a spontaneous tempo, or another alternative. However, at this point, our choice is the first explanation, that participants might be simply feeling the beat for the shorter interval. This choice comes with multiple supports in our data. Tapping intervals of spontaneous tempo documented in the literature are around 600ms, much slower than our frequent tapping rates. Also at a spontaneous rate people’s tap variability is typically very low. But in our data, for sparse tapping the tap variability is high.

Furthermore, in the sparse tapping, inter-tap interval decreased with increases in tempo (thus overall shorter inter-stimulus intervals) as shown in Regression results also in S3 in supplementary materials. This supports the idea that participants still systematically responded to the overall stimulus characteristics, rather than expressing arbitrary randomness, or performing spontaneous (or preferred) tempo tapping.

Finally, if participants’ sense of beat broke down, they could have simply chosen not tapping at all. 

These points are now explicitly articulated in the discussion (p.27-28). 

From the second paragraph and onwards the general discussion is much more to the point and here it also becomes clear that the frequent tapping “strategy” in fact is another name for losing the ability to extract a beat from the stimulus, as is clear from the following: “when stimulus coupling was weaker, people resorted to more frequent tapping, revealing individual differences at which level of coupling to do so. This departure from regular beat tapping indicates that people could no longer rely on the dynamics of the sound onset density change in order to extract beat.» The identification of different breakdown levels for beat extraction for the three groups is a major finding. However, by presenting fast tapping as a strategy rather than inadequate tapping, the authors are obscuring this interesting point. In general, too much attention is given to frequent tapping as such. I would say that the only interesting thing here is at what coupling level the shift happens for the different groups.

Thank you for the comment, and we are happy to hear that the reviewer also thinks the shift is interesting. The question about ‘strategy’ overlaps with the other reviewer. As in our response to the previous remark, we agree with the reviewer that the fast tapping may not constitute a strategy as we do not know if this necessarily is a conscious decision. As such, we have now rephrased as ‘mode’ or ‘approach’ throughout the revised manuscript.

We agree that breakdown levels differing between participants’ category is an important finding. We further propose that the fast tapping possibly reflects sequential processing of multiple stimulus intervals, rather than an absence of regular tapping or ‘inadequate tapping’, as elaborated above. We highlight these arguments in the general discussion (p.28).

5. The structure of the conclusion should be improved. Please revisit your main hypothesis and sub-hypothesis and address the relevant results point by point.

Thank you for this comment. We have modified the conclusion to revisit each of the main hypothesis in the introduction and highlighted the important aspects of the findings (p.33-34). 

Minor points:

p. 4 ‘Sound onset density’ is introduced before the method is presented and it is not immediately clear what this means at this point in the manuscript. Please explain.

Thank you for pointing this out. We added a sentence that defines ‘sound onset density’ (p.4). Also, in response to the other reviewer’s comment, we included a new figure (Figure 1) to show examples of various coupling and schematics of sound onset density with acoustic waveforms. 

p. 4: “In that sense, each individual may serve another agent with a coupling factor, contributing to the whole system.” Unclear, please explain.

We have rephrased the sentence to explain the idea that individuals in a music ensemble also entrain to others, collectively contributing to the governing tempo (p.4). 

p. 5: “This predicts that stronger coupling conditions would result in more consistent tapping intervals and phase relationship to other participants compared to weaker coupling conditions.» Here one might get the impression that several participants are tested simultaneously.

This is a good point also the other reviewer mentioned. We have rephrased the sentence to the following: “This predicts that stronger coupling conditions would result in more consistent tapping intervals and tap-stimulus phase relationships compared to weaker coupling conditions”.

p. 9: “This resulted in a total of 30 audio stimulus files.” I was a bit confused here. Isn’t the correct number 60: 3 coupling strengths x 5 tempi x 4 versions = 60?

Thank you for finding the mistake. We had 2 versions (not 4) for each coupling and tempo condition, resulting in 30. The number is corrected now (p. 9).

p. 9-10: Using the space bar on a laptop is not the best solution for obtaining the millisecond precision needed for research into microtiming. I appreciate that the authors discuss potential limitations related to latency issues and provide the technical specifications of their equipment towards the end of the section “Test Procedure”. However, this part of the manuscript is somewhat difficult to follow and would benefit from more clarity.

Thank you for the comments. These technical notes are now in a separate paragraph, and shortened to improve readability (p. 10). 

“The generative coupled oscillator model produced stimuli using different coupling strengths. to distribute sound onsets different periodic moments in time.” This sentence is not clear and feels incomplete. Please rewrite.

We have rewritten the sentence: “The generative coupled oscillator model produced stimuli by using different coupling strengths to distribute sound onsets around periodic moments in time.” (p.11).

“Briefly, the generated stimuli’s average angle, � at each time step was used to determine beat bins using the beat segmentation algorithm extracted the referent beat centers.” Incomplete sentence. Missing comma after step?

We have rewritten this sentence: “Briefly, the generated stimuli’s average angle, � at each time step, was used to determine beat bins from the beat segmentation algorithm extracted the referent beat centers.” (p.11).

“...circularly mapped onto the beat bins in order to perform phase coherence analysis (per beat region)…” This is an example of beat bin referring to the signal. This should be made clear.

Thank you for the suggestion. Yes, we should make clear that here we refer to the beat window, not the perceptual beat bin. We have rewritten the sentence to the following: “The tapping data for each stimulus sequence were extracted as inter-tap interval (ITI) for each tempo and coupling condition, and circularly mapped onto the beat segments in order to perform phase coherence analysis and to compare with the aforementioned stimuli’s phase coherence.” (p. 11).

«For ITI analysis, we are concerned the amount of time (period)…» Unclear. Please fill in missing word(s).

Thank you for pointing this out. These sentences are rephrased (p. 11).

p. 16: What kind of musical training? Formal or practice or both?

Since we did not provide definition of musical training, the question is taken to be self-interpretative. This information is added for clarification: “Years of musical training (self-interpretative, 43 answered the question)” (p. 15).

p. 27: “…significantly larger for the strong coupling condition compared to the weaker coupling conditions…” Larger or perhaps ‘longer’?

We have changed the wording to ‘longer’ as suggested (p.23). 

p. 31: In the first sentence of the second paragraph of the general discussion (p. 31), the authors state: “Our results confirm that strongly coupled stimuli clearly encourages regular beat extraction in all participants, supporting the idea of ‘beat bin’ (Danielsen, 2010; Danielsen et al., 2019) as we hypothesized.» As this reads now, it is a bit unclear whether “as we hypothesized” refers back to the first or second part of the sentence. It should also be noted here that the beat bin model predicts that narrow bins in the signal, or strongly coupled stimuli, will encourage regular beat extraction in the sense of both stable period synchronization and tight phase synchronization. Wider bins in the signal will produce wider perceptual bins, that is, looser phase synchronization and more tapping variability. As mentioned above, the latter finding is important and should be listed among the main findings.

Thank you, we have revised this and subsequent sections and highlighted that the perceptual bins become wider when looser phase synchronization occurred, indeed in line with the beat-bin model, when people are still successful in extracting regular beats by integrating multiple onsets (p.27). 

p. 32: “The largest intercept and fastest decay rate in the Regular group indicating their taps were settled over a longer time period with more dynamic error correction.” Correct the syntax. It should also be noted here that this could be because of a general inclination towards more narrow beat bins in this group.

We have changed the sentence to: “The largest intercept and fastest decay rate in the Regular group indicates their taps were settled over a longer time period with more dynamic error correction” (p.29).

Regarding the question if the Regular group has narrower beat bins, we need further consideration. Our data show that the Regular group was most tolerant against the temporal spread of multiple onsets, because they were able to still feel the beat despite the spread. As such, their beat bin size should be actually wider than the other two groups. Further, such a high tolerance and a wide beat bin size might relate to their slow adaptation processes, where they have to adjust and align the internal beat intervals (relative to the allowance of the adjustment) more gradually. This idea is now mentioned in the discussion (p.29).

“…if people have different strategies in approaching towards the uncertain and various timing intervals…” When approaching or In their approach to.

Regarding the possibility of different strategies, it might be that the wider perceptual beat bins of the hybrid group facilitate stable periodicity because more taps will be experienced as within the beat bin when the bins are wide. See Matthews et al. 2022 for an interesting article on the relationship between perceived and measured synchrony.

Thank you for this comment and we have modified the sentence with the proper expression. 

We agree that one way to explain the hybrid group’s response to weaker coupling is that they maintain wider perceptual beat bins, related to the discussion point above. This point is now mentioned in the discussion (p.29). 

Also, thank you for suggesting this article, which is now cited in this section (p.29).

p. 33: “Altogether, our paradigm provides a new way to assess individual tapping strategies and adds to the current knowledge of beat induction processes that are informed by top-down predisposition/preferences and bottom-up stimulus-driven processes.» How? Please unpack this claim. 

We have reorganized this paragraph and this sentence is removed. We still mention the influences of prior experience and individual preference in musical beat and metric grouping (p. 30).

In the discussion, it is sometimes unclear whether you refer to other studies or your own work. Be sure to make this clear by adding “previous research has shown… “ etc.

Wherever applicable, we used phrases such as ‘our results’, ‘our paradigm’ etc. 

We thank Reviewer #1 for the close reading and detailed and constructive suggestions.

Reviewer #2: I read this article with much interest Interindividual differences in tapping to auditory stimuli with various timing properties. The main finding that participants use different strategies to coordinate their movement with more or less salient beats is very interesting and contributes to our understanding of humans’ rhythm processing skills.

Thank you for your interest and positive evaluation of our study. 

The writing of the paper should be improved. There are frequent grammatical mistakes (e.g., “Our results confirm that strongly coupled stimuli clearly encourages”, p.31). Numerous examples of unclear sentences can be found throughout the manuscript. I strongly encourage the authors to have the article proofread by several people. Also, the length of the paragraphs (almost 1.5 pages for some of them in the discussion section) is too long, making it particularly difficult to follow. For example, page 34 should be divided into three paragraphs, focusing on beat deafness, spontaneous production rates, and neuronal bases, respectively.

Thank you for the suggestion. In this revision, whenever applicable we paid attention to the paragraph length, correctness, and clarity. We also have asked our colleagues for proofreading and editing. 

We have split the aforementioned paragraph into three separate sections as suggested (p.31-32). 

p5, §2 “to other participants” (towards the bottom of the page): is that correct? This suggests that this a question address by a cross subject design

This was also mentioned by Reviewer #1. We have removed the phrase from the sentence (p. 6). 

p6, §1 “might employ different behavioral patterns, related to their internal tendencies

that interact with external sounds of different coupling strength” (towards the top the page): it is not clear what behavioral patterns and internal tendencies mean here, and it does not seem that they are explicitly explained earlier in the introduction.

Thank you for this comment. We have removed the phrase ‘internal tendencies’ and revised the sentence to explain the idea that certain people may focus on different auditory features in order to extract a sense of beat from sequences with less well-defined beat centers (p.6).

There are some points that are unclear to me in the hypotheses formulated. It should be specified that coupling, here, refers to the coupling between the different oscillators generating the sounds. It is particularly important because it is not easy to understand that the stimuli consisted in clustered sounds with a “beat” at the peak level metronome sequences (i.e., single sounds) at a quasi-periodic frequency. This should be clarified in the introduction. It would help to include a schematic of the stimuli in the strong, medium, weak, and none phase coherence conditions:

Strong: I I I I I I

Weak: I I I I I I I I

It becomes clearer towards the results and discussion sections stimuli are actually clustered sounds. Individual dots representing the sound of each oscillator could be added above the bars.

I would appreciate if the authors could provide example stimuli in the next submission.

Thank you for this suggestion. In the introduction, we added a new diagram (now Figure 1) that shows examples of how the oscillators in the generative model translate to the stimulus acoustic waveforms with different couplings. The body text also includes added explanation (p.7) . 

Experiment 1

I am not convinced that presenting this first experiment is necessary. It is not common to present pilot experiments, and the added value here seems very small. The article is dense, with very complex analysis. I think it would be much easier to read, and the main results will be conveyed more efficiently if this experiment is not presented. The authors can mention in the introduction that a short-scale, in-person pilot study using a smaller range of stimuli tempi and duration was conducted prior to the experiment to show that their design capture people’s naturalistic tapping behavior.

Thank you for this comment. We considered the possibility but ultimately decided to retain Exp. 1 in the current revision. While the subsequent experiment had a much larger subject size with a more substantive in analysis, it was performed entirely online; we believe that the in-person testing in Exp. 1 with a small cohort —which consistently found stronger coupling facilitating beat entrainment and the emergence of ‘frequent tapping’- later confirmed by Exp. 2 -- strengthens the claims we draw altogether. Still, the reviewer has a valid point for the readability. We made edits wherever possible to improve the clarity throughout the manuscript. 

Experiment 2

Why did the authors use the nITI instead of the mean asynchrony, or the vector angle? I understand after reading the discussion that it is because the relative timing is not reliable with the experimental setting. This should be explained earlier in the article. Note that this increases significantly the complexity of the analyses.

There are some repetitions across the methods and the results section, especially for the clustering analyses.

Please report effect sizes for statistical analyses when relevant.

Thank you for this comment. 

For the vector angle, please see our results in the ‘Phase Coherence Analysis” for Exp.1 and Exp. 2. 

We made the technical notes about the constraint (explained in Exp.1 method section) much shorter and concise, and explicitly explained that the latency did not allow us to compute mean asynchrony (p. 10). We removed the repetitions as much as possible. We also included the effect size (generalized eta squared in ANOVA results) when applicable.

The results second paragraph of page 24 (beginning with “we examined”) could be summarized in a table to reduce the text length

Thank you for this suggestion. We have retained the paragraph as is, because it turned out making a new table did not save the space, but increase the overall length in this section.

Page 27: the wording of the presentation of the ANOVA is awkward (The main effect (…) was significant because the post-hocs show)

We have rephrased the text sentences in the ANOVA description wherever applicable (p. 23-24).

Also, why was the Group not included as an independent variable in the ANOVA, instead of running the same anova on the 3 different groups? I expect an overall difference between the 3 groups by eyeballing figure 4A

Thank you for the question. We did not include Group assignment as a factor of ANOVA to avoid the tautology, because group designation was made through clustering of nITI (which is the dependent variable of the ANOVA). 

The order of the figures is confusing. For example, figure 3C is presented after figure 4 in the text.

Thank you for pointing this out. Figure 5 (formerly Figure 4) shows the results of the exponential curve fitting from the SD of each subject group’s time course analysis which is derived from the nITI results. As such we felt as though it proceeds more naturally to summarize this section just after presenting the nITI results rather than returning to it after presenting the Phase Coherence analysis illustrated in Figure 4C (formerly Figure 3C).

Page 28, paragraph 3 (beginning with “Interestingly”). What analyses this interpretation correspond to? Are they supported by statistical analyses?

Thank you for the question. To avoid the confusion, we have made this paragraph as a continued part of the previous paragraph describing the phase coherence analysis. Phase coherence magnitude is already an aggregate, statistic value. The distributions that makeup the phase coherence were already compared by the circular statistics (Rayleigh) mentioned in the previous part: all phase angles (tap and stimuli) were considered significant with the exception of the ‘none’ coupled stimulus (p. 24). 

In general, the authors could be more cautious with interpretation of results not supported by inferential statistics (see also p25, last paragraph), or at least explain better why these interpretations are performed. Again, there are a lot of results and analyses that are very difficult to grasp for the reader. Focusing on the main findings would definitely help.

Thank you for this comment. Similarly, to the reason behind ANOVAs without group as a factor as discussed in the above response, for the results about the proportion of the nITI clusters, performing any inferential statistical comparison should be avoided, because these data provided the criteria for classifying participants into group, This is now explained in the section (p.25). Despite our initial drafting, we recognize the importance of the reviewer's suggestion to improve the clarity for readers. Therefore, we have revised the manuscript to ensure that our text is presented more clearly throughout.

The main findings of the group characterization are definitely our focus. It is important to emphasize that our groups are defined solely by their tapping pattern differences, rather than screening criteria or demographic differences. As such, we think that it is important to systematically examine the various aspects of the tapping data and provide supporting evidence for group characterization.

Discussion

“Frequent tapping strategy” does not seem like an appropriate way to say that people tap faster than stimulus tactus.

This comment overlaps with Reviewer #1. We think that the word ‘strategy’ may imply that it is a conscious choice. We have revised the wording to ‘mode’ or ‘approach’. Please see our responses to the comments from Reviewer #1 above. The relevant issues are now discussed in the General Discussion section (p. 27-28).

Regarding spontaneous production rates. A recent paper showed the influence of people’s spontaneous rates of production in the context of non-linear dynamic modeling (Bégel, Demos, Wang, & Palmer, 2022).

Thank you for the reference. We have incorporated this article in the general discussion (p. 31). 

“tracking the collective rhythm rather than tracking separate streams” (p31, first paragraph). The authors should clarify what they mean by collective rhythm.

Thank you for this comment. The wording ‘collective rhythm’ has been rephrased as ‘surface level rhythm’ (p. 28). 

Relatedly, in the immediate preceding section, we have introduced the term ‘integrative mode’ to aggregate onsets to extract common beats versus a ‘sequential mode’ to track surface rhythms formed by the subsequent onsets. (p.28).Page 33, middle paragraph. I don’t understand the argumentation. It seems to me that the paragraph covers different points, and it is not clear how they are related to the study, especially because these points are not presented in the introduction. I also miss the general idea here. What does “short stimulus onset intervals” mean? I understand that it’s the interval between each oscillator, but this is very confusing.

Thank you for this comment. The paragraph in question now appears in page 30, starting with a new sentence (“What underlies the sparse tapping behavior?”). We rephrased this entire paragraph to make it much shorter with a clearer argument in speculating the nature of the sparse tapping with relevant literature about tracking or mimicking behavior (p. 30).

Try to shorten the paragraph, anything that might be sidetracking. Needs to be essential to stay in. 

Most part of the general discussion has been rewritten and/or shortened. In response to this suggestion, we also made much more brief remarks for the final paragraph (just before conclusion) (p. 32).

Minor comments

When referring to humans, “Female” should be employed as an adjective. Although the formulation used by the authors (5 females) is frequent in scientific articles, my opinion is that another noun should be used instead (women), or female should be employed as an adjective, as it can be perceived as offensive (the definition of Wikipedia clearly states “now possibly offensive in certain contexts”). I am not a native English speaker, but it seems to me that using the singular form of female in the sentence would turn it into an adjective implicitly related to the word “volunteers” and therefore be grammatically correct: “10 volunteers from the Stanford University community took part in this study (5 female, M age = 31 years, SD = 5.6 years, all right handed).”

Thank you for pointing this out. We have modified this term by following the editor’s suggestion (p. 7).

It also seems to me that contracted forms (“It’s”, p8 §1) should not be used

We removed any contractive form of “It is” throughout the manuscript. 

We thank Reviewer #2 for the detailed and constructive suggestions.

---

## [Decision Letter · Decision Letter 1]

15 Jun 2023

PONE-D-22-25174R1Individual differences of limitation to extract beat from Kuramoto coupled oscillators: transition from beat-based tapping to frequent tapping with weaker couplingPLOS ONE

Dear Dr. Nolan Lem,

Thank you for submitting your manuscript to PLOS ONE. After careful consideration, we feel that it has merit but does not fully meet PLOS ONE’s publication criteria as it currently stands. Therefore, we invite you to submit a revised version of the manuscript that addresses the points raised during the review process.

We look forward to receiving your revised manuscript.

Kind regards,

Tiago Pereira

Academic Editor

PLOS ONE

Journal Requirements:

Reviewers' comments:

Reviewer's Responses to Questions

**Comments to the Author**

1. If the authors have adequately addressed your comments raised in a previous round of review and you feel that this manuscript is now acceptable for publication, you may indicate that here to bypass the “Comments to the Author” section, enter your conflict of interest statement in the “Confidential to Editor” section, and submit your "Accept" recommendation.

Reviewer #1: All comments have been addressed

Reviewer #2: All comments have been addressed

2. Is the manuscript technically sound, and do the data support the conclusions?

Reviewer #1: (No Response)

Reviewer #2: Yes

3. Has the statistical analysis been performed appropriately and rigorously? 

Reviewer #1: (No Response)

Reviewer #2: Yes

4. Have the authors made all data underlying the findings in their manuscript fully available?

Reviewer #1: (No Response)

Reviewer #2: Yes

5. Is the manuscript presented in an intelligible fashion and written in standard English?

Reviewer #1: (No Response)

Reviewer #2: Yes

6. Review Comments to the Author

Reviewer #1: The manuscript reads much better and all my comments and concerns have been addressed. The study design, results and implications are much more clearly communicated. The work will be an excellent contribution to the research field.

Below is a list of suggestions for minor revisions/further improvements:

- The abstract needs to be rewritten to improve clarity and better reflect the revisions that have been made to the main manuscript.

p. 3 “sound sequences that varied the degree of such deviations systematically”. I would consider using “spread in multiple sound onsets” instead of deviations here (it is not clear what the deviations deviate from).

p. 3 I suggest changing the following sentence in the second paragraph to: “For example, listeners’ tolerance for pulse attribution is nearly twice as large as their threshold for detecting simple irregularity…”

p. 5, line 9: Change “not only the” to “is not the only”

p. 5, last paragraph, first line: Change “which” to “with”. This sentence is generally a bit difficult to parse.

p. 6: The third hypothesis could be more clearly formulated. The mention of auditory features is particularly confusing (auditory features are not discussed anywhere else in the manuscript).

p. 13, line 2: Perhaps say participants instead of group since the “group” consists of only 2 people �

p. 17-19. Parts of the text under “Data analysis” on p. 17 is repeated on pages 18 and 19.

p. 18, lines 3-4: Consider replacing “frequent tapping responses” with “high frequency tapping responses” (the former could be misunderstood)

p. 19, second paragraph: Change the following sentence to: “In the Fast tapping group, the majority of tap responses for the three weaker coupling conditions belongs to the frequent tapping cluster, increasing from 85% for the medium, to 94% for the weak, to 99%... “

p. 21 second paragraph – make clear that the shorter ITIs is mainly driven by a switch to frequent tapping? (it’s probably not a gradual thing)

p. 25, first paragraph: characterized by (not characterized with)

and “switched” is not a good word here. Consider rephrasing.

p. 25, second paragraph, first line: highly frequent tapping (not very frequent)

p. 26, lines 3-4: The sentence starting with “Most importantly, … “ is not clear. Please rephrase.

p. 26, second paragraph, line 3: What kind of underpinning?

p. 26-27: It’s sometimes unclear what you mean by beat window – please define or repeat if defined earlier on in the manuscript.

p. 28, second paragraph (references). The following study shows that prior experiences (expert stylistic knowledge) influences beat bins and beat perception in general.

Danielsen, A., Nymoen, K., Langerød, M. T., Jacobsen, E., Johansson, M., & London, J. (2021). Sounds familiar (?): Expertise with specific musical genres modulates timing perception and micro-level synchronization to auditory stimuli. Attention, Perception, & Psychophysics, 1-17.

p. 28, end of last complete paragraph: “ a motor process”

p. 29, second line: consider adding “…easily in all the conditions with some degree of coupling.”

Reviewer #2: Thank you for carefully adrressing my comments. I just noticed a mistake in one reference (J. A. Grahn, p5)

7. PLOS authors have the option to publish the peer review history of their article (what does this mean?). If published, this will include your full peer review and any attached files.

Reviewer #1: **Yes: **Anne Danielsen

Reviewer #2: **Yes: **Valentin Bégel

---

## [Author Response · Author response to Decision Letter 1]

8 Jul 2023

Dear reviewers,

We appreciate your detailed and insightful comments. We think that our manuscript has improved by taking into account your criticisms and suggestions. Below we respond to each of your comments and note where and how those responses are handled in the revised manuscript. Revisions to the manuscript and our responses to comments below are in green font. 

Reviewer #1: The manuscript reads much better and all my comments and concerns have been addressed. The study design, results and implications are much more clearly communicated. The work will be an excellent contribution to the research field.

Below is a list of suggestions for minor revisions/further improvements:

- The abstract needs to be rewritten to improve clarity and better reflect the revisions that have been made to the main manuscript.

Thank you for this comment. We have revised the abstract to better accommodate the revisions in this manuscript. 

p. 3 “sound sequences that varied the degree of such deviations systematically”. I would consider using “spread in multiple sound onsets” instead of deviations here (it is not clear what the deviations deviate from).

Thank you for this suggestion. We have changed the wording as suggested. 

p. 3 I suggest changing the following sentence in the second paragraph to: “For example, listeners’ tolerance for pulse attribution is nearly twice as large as their threshold for detecting simple irregularity…”

We have changed the wording as suggested

p. 5, line 9: Change “not only the” to “is not the only”

We have changed the wording as suggested

p. 5, last paragraph, first line: Change “which” to “with”. This sentence is generally a bit difficult to parse.

We have changed the wording as suggested

p. 6: The third hypothesis could be more clearly formulated. The mention of auditory features is particularly confusing (auditory features are not discussed anywhere else in the manuscript).

Thank you for pointing this out. We removed any reference to auditory features and we added “When confronted with multiple inter-onset intervals” to better specify the stimulus characteristic we are alluding to. 

p. 13, line 2: Perhaps say participants instead of group since the “group” consists of only 2 people �

Thanks for this comment, we have changed the wording to ‘participants’. 

p. 17-19. Parts of the text under “Data analysis” on p. 17 is repeated on pages 18 and 19.

Thank you for pointing out the potential redundancy. However, we could not clearly identify the parts in question. We made a few wording changes in p. 18-19 for clarity. We would appreciate if further changes are necessary.

p. 18, lines 3-4: Consider replacing “frequent tapping responses” with “high frequency tapping responses” (the former could be misunderstood)

We changed the wording as suggested.

p. 19, second paragraph: Change the following sentence to: “In the Fast tapping group, the majority of tap responses for the three weaker coupling conditions belongs to the frequent tapping cluster, increasing from 85% for the medium, to 94% for the weak, to 99%... “

We changed the wording as suggested.

p. 21 second paragraph – make clear that the shorter ITIs is mainly driven by a switch to frequent tapping? (it’s probably not a gradual thing)

Thank you for noticing an interesting potential connection here. The observation that the within-segment mean nITI in the Regular group is approaching to 0.75 on average for later tapping sections (Figure 4B, Regular group) in the none-coupling condition indeed coincided with the observation that the longest nITI in the cluster 3 (frequent tapping) which is around 0.8 in the none-coupling condition (the area of R < 0.2, Figure 3). 

But in this ANOVA result section, the dependent variable is the mean and SD of nITIs within a short 4-tap segment (over the 1st to 7th segement from the beginning), different from the whole trial sequence average we used for clustering. This means that we do not have moment-to-moment representation of frequent tapping in the current data. We agree with the reviewer that it would be exciting to elucidate the shift occurs gradually or suddenly when we have the next opportunity to explore this particular question. 

As such, we have decided not to add further interpretation in the ANOVA results section to avoid introducing unnecessary confusion about two different metrics. Nonetheless, the interpretation about this piece of result is mentioned in the later discussion (p. 27). 

p. 25, first paragraph: characterized by (not characterized with)

and “switched” is not a good word here. Consider rephrasing.

Thank you for this comment. We have rephrased the sentence using the word ‘transition’ in its place. 

p. 25, second paragraph, first line: highly frequent tapping (not very frequent)

We changed the wording as suggested.

p. 26, lines 3-4: The sentence starting with “Most importantly, … “ is not clear. Please rephrase.

We changed the wording as suggested.

p. 26, second paragraph, line 3: What kind of underpinning?

Thank you for this comment. We have clarified what we mean by ‘underpinning’ here. 

p. 26-27: It’s sometimes unclear what you mean by beat window – please define or repeat if defined earlier on in the manuscript.

We now rephrased it with ‘beat interval’ in p. 26-27.

p. 28, second paragraph (references). The following study shows that prior experiences (expert stylistic knowledge) influences beat bins and beat perception in general.

Danielsen, A., Nymoen, K., Langerød, M. T., Jacobsen, E., Johansson, M., & London, J. (2021). Sounds familiar (?): Expertise with specific musical genres modulates timing perception and micro-level synchronization to auditory stimuli. Attention, Perception, & Psychophysics, 1-17.

Thank you. This reference has been added. 

p. 28, end of last complete paragraph: “ a motor process”

Thank you. This has been changed. 

p. 29, second line: consider adding “…easily in all the conditions with some degree of coupling.”

Thank you for this comment. We have updated the sentence with the suggested phrase. 

Reviewer #2: Thank you for carefully adrressing my comments. I just noticed a mistake in one reference (J. A. Grahn, p5)

Thank you for finding the reference typo. We corrected a few other mistakes from the reference manager software glitches.

---

## [Decision Letter · Decision Letter 2]

8 Aug 2023

PONE-D-22-25174R2Individual differences of limitation to extract beat from Kuramoto coupled oscillators: transition from beat-based tapping to frequent tapping with weaker couplingPLOS ONE

Dear Dr. Lem,

Thank you for submitting your manuscript to PLOS ONE. The referee has suggested some minor revisions that will improve the manuscript. So, we also recommend you, please implement them. 

Kind regards,

Tiago Pereira

Academic Editor

PLOS ONE

Journal Requirements:

Reviewers' comments:

Reviewer's Responses to Questions

**Comments to the Author**

1. If the authors have adequately addressed your comments raised in a previous round of review and you feel that this manuscript is now acceptable for publication, you may indicate that here to bypass the “Comments to the Author” section, enter your conflict of interest statement in the “Confidential to Editor” section, and submit your "Accept" recommendation.

Reviewer #1: All comments have been addressed

2. Is the manuscript technically sound, and do the data support the conclusions?

Reviewer #1: (No Response)

3. Has the statistical analysis been performed appropriately and rigorously? 

Reviewer #1: (No Response)

4. Have the authors made all data underlying the findings in their manuscript fully available?

Reviewer #1: (No Response)

5. Is the manuscript presented in an intelligible fashion and written in standard English?

Reviewer #1: (No Response)

6. Review Comments to the Author

Reviewer #1: Thank you for carefully addressing my comments and suggestions. I have a few more minor comments/suggestions that I think would improve the paper. Given that these are addressed, I am happy to recommend the paper for publication and do not need to see another version.

Abstract:

The new abstract better reflects the content of the article. However, make sure that the wording matches the text in the main manuscript (for example, use «highly frequent tapping» instead of «frequently tapped»). The use of the word «while» is also somewhat confusing. I also suggest asking a native speaker to do a copy edit of the abstract.

P. 28: «Such criteria might be further informed by different prior experiences with or individual preferences for musical beats or metrical groupings (Jongsma, Desain, and Honing 2004; Snyder, Pasinski, and McAuley 2011; Danielsen et al. 2022).»

Nori Jacoby’s large-scale cross-cultural study of metrical priors would give further support to this point.

Reference:

Jacoby, N., Polak, R., Grahn, J., Cameron, D. J., Lee, K. M., Godoy, R., ... & McDermott, J. H. (2021). Universality and cross-cultural variation in mental representations of music revealed by global comparison of rhythm priors.

P. 31:

«Our results also confirmed our first hypothesis in that when the coupling is stronger, people show integrative processing of multi-onset information to achieve regular tapping which becomes more variable, consistent with the gradual spread of the perceptual beat-bin.»

Something missing after «becomes more variable» in this sentence? Perhaps change to «…becomes more variable with weaker coupling…»? I also suggest replacing «spread» with «widening».

References: I noticed some inconsistencies in the style of the references. Make sure that all references are complete, consistently styled, and in line with the journal’s requirements.

7. PLOS authors have the option to publish the peer review history of their article (what does this mean?). If published, this will include your full peer review and any attached files.

Reviewer #1: **Yes: **Anne Danielsen

---

## [Author Response · Author response to Decision Letter 2]

21 Aug 2023

Dear reviewers,

We appreciate your detailed and insightful comments. We think that our manuscript has improved by taking into account your criticisms and suggestions. Below we respond to each of your comments and note where and how those responses are handled in the revised manuscript. Revisions to the manuscript and our responses to comments below are in blue font. 

Reviewer #1: Thank you for carefully addressing my comments and suggestions. I have a few more minor comments/suggestions that I think would improve the paper. Given that these are addressed, I am happy to recommend the paper for publication and do not need to see another version.

Thank you for all your very insightful comments and we are pleased to hear you will recommend the paper for publication. 

Abstract:

The new abstract better reflects the content of the article. However, make sure that the wording matches the text in the main manuscript (for example, use «highly frequent tapping» instead of «frequently tapped»). The use of the word «while» is also somewhat confusing. I also suggest asking a native speaker to do a copy edit of the abstract.

Thank you for the suggestion. We rephrased the wordings in those places.

P. 28: «Such criteria might be further informed by different prior experiences with or individual preferences for musical beats or metrical groupings (Jongsma, Desain, and Honing 2004; Snyder, Pasinski, and McAuley 2011; Danielsen et al. 2022).»

Nori Jacoby’s large-scale cross-cultural study of metrical priors would give further support to this point.

Reference:

Jacoby, N., Polak, R., Grahn, J., Cameron, D. J., Lee, K. M., Godoy, R., ... & McDermott, J. H. (2021). Universality and cross-cultural variation in mental representations of music revealed by global comparison of rhythm priors.

Thank you for this suggestion. We have added this reference in the manuscript in the appropriate place.

P. 31:

«Our results also confirmed our first hypothesis in that when the coupling is stronger, people show integrative processing of multi-onset information to achieve regular tapping which becomes more variable, consistent with the gradual spread of the perceptual beat-bin.»

Something missing after «becomes more variable» in this sentence? Perhaps change to «…becomes more variable with weaker coupling…»? I also suggest replacing «spread» with «widening».

Thank you for the suggestion. We rephrase the few sentences here to include the suggested wording for clarification (now p.41)

References: I noticed some inconsistencies in the style of the references. Make sure that all references are complete, consistently styled, and in line with the journal’s requirements.

Thank you for this suggestion. We have edited the sentence to provide more clarity. We have also changed the format of the manuscript’s citations to abide by the Vancouver citation format suggested by PLOS ONE and solved the reference managing software’s technical problems. Each citation was checked for its inclusion in the references section.

---

## [Editor Report · Decision Letter 3]

12 Sep 2023

Individual differences of limitation to extract beat from Kuramoto coupled oscillators: transition from beat-based tapping to frequent tapping with weaker coupling

PONE-D-22-25174R3

Dear Prof. Lem,

We’re pleased to inform you that your manuscript has been judged scientifically suitable for publication and will be formally accepted for publication once it meets all outstanding technical requirements.

Kind regards,

Tiago Pereira

Academic Editor

PLOS ONE
---

## [Editor Report · Acceptance letter]

28 Sep 2023

PONE-D-22-25174R3 

Individual differences of limitation to extract beat from Kuramoto coupled oscillators: transition from beat-based tapping to frequent tapping with weaker coupling 

Dear Dr. Lem:

I'm pleased to inform you that your manuscript has been deemed suitable for publication in PLOS ONE. Congratulations! Your manuscript is now with our production department. 

Kind regards, 

on behalf of

Dr. Tiago Pereira 

Academic Editor

PLOS ONE